# Artificial selection reveals complex genetic architecture of shoot branching and its response to nitrate supply in Arabidopsis

Hugo Tavares[1☉], Anne Readshaw[2☉], Urszula Kania[1], Maaike de Jong[1], Raj K. Pasam[1], Hayley McCulloch[1], Sally Ward[1,2], Liron Shenhav[1], Elizabeth Forsyth[1], Ottoline Leyser[1,2]*

1 Sainsbury Laboratory, University of Cambridge, Cambridge, United Kingdom, 2 Department of Biology, University of York, York, United Kingdom

☉ These authors contributed equally to this work.
* ol235@cam.ac.uk

## Abstract

Quantitative traits may be controlled by many loci, many alleles at each locus, and subject to genotype-by-environment interactions, making them difficult to map. One example of such a complex trait is shoot branching in the model plant Arabidopsis, and its plasticity in response to nitrate. Here, we use artificial selection under contrasting nitrate supplies to dissect the genetic architecture of this complex trait, where loci identified by association mapping failed to explain heritability estimates. We found a consistent response to selection for high branching, with correlated responses in other traits such as plasticity and flowering time. Genome-wide scans for selection and simulations suggest that at least tens of loci control this trait, with a distinct genetic architecture between low and high nitrate treatments. While signals of selection could be detected in the populations selected for high branching on low nitrate, there was very little overlap in the regions selected in three independent populations. Thus the regulatory network controlling shoot branching can be tuned in different ways to give similar phenotypes.

**Data Availability Statement:** Raw Illumina sequence data are available from NCBI's Short Read Archive (BioProject accession PRJNA611751). All other data and analysis scripts

## Author summary

Unlike in animals, plant development happens continuously throughout an individual's lifespan and strongly interacts with the external environment. An example is the regulation of shoot branching, which depends on the regulated activity of buds in the axils of leaves. The switch between dormancy and activity in buds involves the integration of both internal (hormonal) and external (environmental) signals. We have previously shown that shoot branching in Arabidopsis is a highly plastic trait in response to nitrate in the soil. This plasticity is genetically variable, however previous studies have failed to elucidate fully its genetic architecture (number, location and effect of the underlying loci). Here, we use an artificial selection approach, combined with genome sequencing and simulations, to tackle this challenge. Our results suggest that this trait is controlled by at least tens of loci of small effect. We show that different populations achieve a similar average number

are available from the Cambridge Apollo Repository: https://doi.org/10.17863/CAM.96047. Analysis scripts, with instructions on how to run them, are available from: https://github.com/tavareshugo/publication_Tavares2023_Nselection.

**Funding:** This work was funded by: a Marie Currie Fellowship (PIEF-GA-2009-252761) from the European Commission (https://ec.europa.eu/research/fp7/index_en.cfm) to MdJ; research grants to OL from the Gatsby Charitable Foundation (GAT3395/PR1 and GAT3071) (https://www.gatsby.org.uk/) and the European Research Council (No. 294514 – EnCoDe) (https://ec.europa.eu/research/fp7/index_en.cfm). The funders had no role in study design, data collection and analysis, decision to publish, or preparation of the manuscript.

**Competing interests:** The authors have declared that no competing interests exist.

of branches, despite selection for different loci, suggesting some genetic redundancy, such that multiple genetic makeups can result in similar phenotypes. Also, the genetic architecture differed between populations selected with high vs low nitrate concentrations in the soil, suggesting that different regulatory pathways may be involved in the activation of dormant buds depending on the nutrient levels available to the plant. Our study thus clarifies the complex interactions between the environment and genetics for shoot branching regulation in plants.

## Introduction

The study of genetic variation of quantitative traits is essential for understanding phenotypic diversity and has important applications, from the study of disease in humans to improvements in crop breeding [1,2]. With the wider availability of genomic data, the last decade has seen an increase in studies aimed at pinpointing loci associated with such traits (quantitative trait loci, QTL) in a variety of organisms. However, the complex genetic architecture of many traits poses challenges in terms of the statistical power to detect such QTL [1,3,4]. The genetic control of a trait may involve many loci, each having a small effect, with some alleles being rare in the mapping populations. In addition, allelic diversity at each locus, population structure, and the effect of environment on the traits under study all complicate the analysis [5–9]. However, well-powered analysis, with ever-increasing sample sizes has been fruitful at uncovering the genetics of even highly polygenic traits [10]. Height in humans is an excellent example, where current sample sizes in the millions have uncovered thousands of loci explaining a large fraction of the heritability in this trait [11].

An alternative to the study of complex trait variation is to use experimental evolution and/or artificial selection to assess how genetically diverse populations respond to selection [12,13]. Responses to selection imply that there is heritable variation for the trait in the population. Additionally, if combined with whole-genome sequencing, new QTL may be identified by scanning the genome for selection signals [13,14]. Because allele frequency changes are the main metric of interest in these "evolve-and-resequence" experiments, they can be made cost-effective by sequencing pools of individuals (Pool-seq), rather than each individual separately [15]. This approach has been mainly popularised in *Drosophila*, where it has been successfully used to dissect complex traits, often involving hundreds of loci (e.g. [16–18]). There are fewer examples of this approach being used in plants, likely due to the logistic challenges involved in growing and breeding them under controlled environments, but it has been successfully used, for example in selection for flower size in *Mimulus* [19], to identify signals of selection in domesticated bean species compared to ancestral relatives [20] and in long-term selection breeding for oil and protein content in corn kernels [21]. Artificial selection experiments also offer the possibility of controlling particular environmental variables to understand how the response of a trait to selection is conditional on the treatment [22]. This makes it possible to address the issue of whether phenotypic plasticity evolves as a correlated response to selection on a particular trait in different environments.

Here, we combine artificial selection and Pool-seq to study the genetics of shoot branching variation in the model plant Arabidopsis. We have previously described substantial genetic variation for this trait, both in natural accessions, and in a sample of multi-parent recombinant inbred (MAGIC) lines [23]. Furthermore, we have shown this trait to be highly responsive to nitrate availability in the soil, but with differences in responses (plasticity) across genotypes, resulting in substantial variation in genotype-by-environment (GxE) interaction. This

variation strongly affects fruit number in different environments, with potential consequences for fitness. However, we were unable to map loci underpinning this variation in natural accessions, and recovered only a modest number of QTL in the MAGIC lines. This led us to hypothesise that a substantial proportion of the heritability of this trait was unaccounted for due to a complex underlying genetic architecture. This is perhaps unsurprising given the known multiplicity of factors that influence shoot branching, including a network of interacting plant hormones which act through multiple targets to mediate their effects on branching [24]. Interestingly, we found that natural variation in branching and its plasticity correlates with other traits, particularly flowering time, implicating even wider genetic networks in the regulation of branching traits [23].

Here, we build on our previous work, by conducting an artificial selection experiment for shoot branching under different nitrate conditions. We used the MAGIC lines described above as founders for replicate populations subjected to different selection regimes. Selection was imposed for 10 generations with different populations grown under nitrogen-limiting or nitrogen-sufficient conditions, which allowed us to assess further the effects of selection conditional on this important nutrient [25]. The results show that shoot branching is heritable in these populations and responsive to directional selection for increased branching, but not stabilising selection for mean branch numbers. Using whole-genome Pool-seq, we scanned the genomes of these populations to identify putative loci that responded to selection. The results were contrasting between low (LN) and high (HN) nitrate conditions, despite their similar heritability estimated from responses to selection. On LN we found several putative selected loci, spread across the genome, with very few overlaps between replicate populations. Simulations showed that this would be expected from a trait controlled by tens of loci with low-effect alleles. On HN populations, however, we found few clear signals, suggesting a different genetic architecture for the trait under this treatment, with our simulations showing that a higher number of loci controlling the trait under these conditions is compatible with this result. Furthermore, we describe the effects of seasonal conditions on our trait, revealing a further interaction between this trait and environmental inputs of relevance to flowering time. We find that directional selection for shoot branching has correlated effects on its plasticity, and on flowering time, providing evidence for a common genetic basis for these traits, which could be further linked to seasonality. Our results therefore reveal new aspects of the genetic architecture of shoot branching and its plasticity that were not detected in standard QTL mapping experiments.

## Results

### Shoot branching increases in response to directional selection

To investigate the effects of artificial selection on shoot branching mean and variance, we designed an experiment using two selection regimes: directional selection for increased branching and stabilising selection against extremes. Furthermore, to determine how the responses to selection differed in different environmental conditions, we included nitrate sufficient (high N, HN) and nitrate deficient (low N, LN) regimes. These treatments were applied by a weekly feed containing 9mM or 1.8 mM $NO_3^-$, respectively (see methods). These treatments were chosen because we have previously shown substantial genetic variation for shoot branching plasticity in Arabidopsis in response to nitrate, suggesting that the best performing genotypes might differ between these two conditions [23].

The populations subjected to selection were founded from a set of ~400 Arabidopsis MAGIC lines [26] (Figs 1A and S1). These inbred lines were randomly paired and crossed, forming identical populations of ~200 F1 founders for each selection regime. Due to the

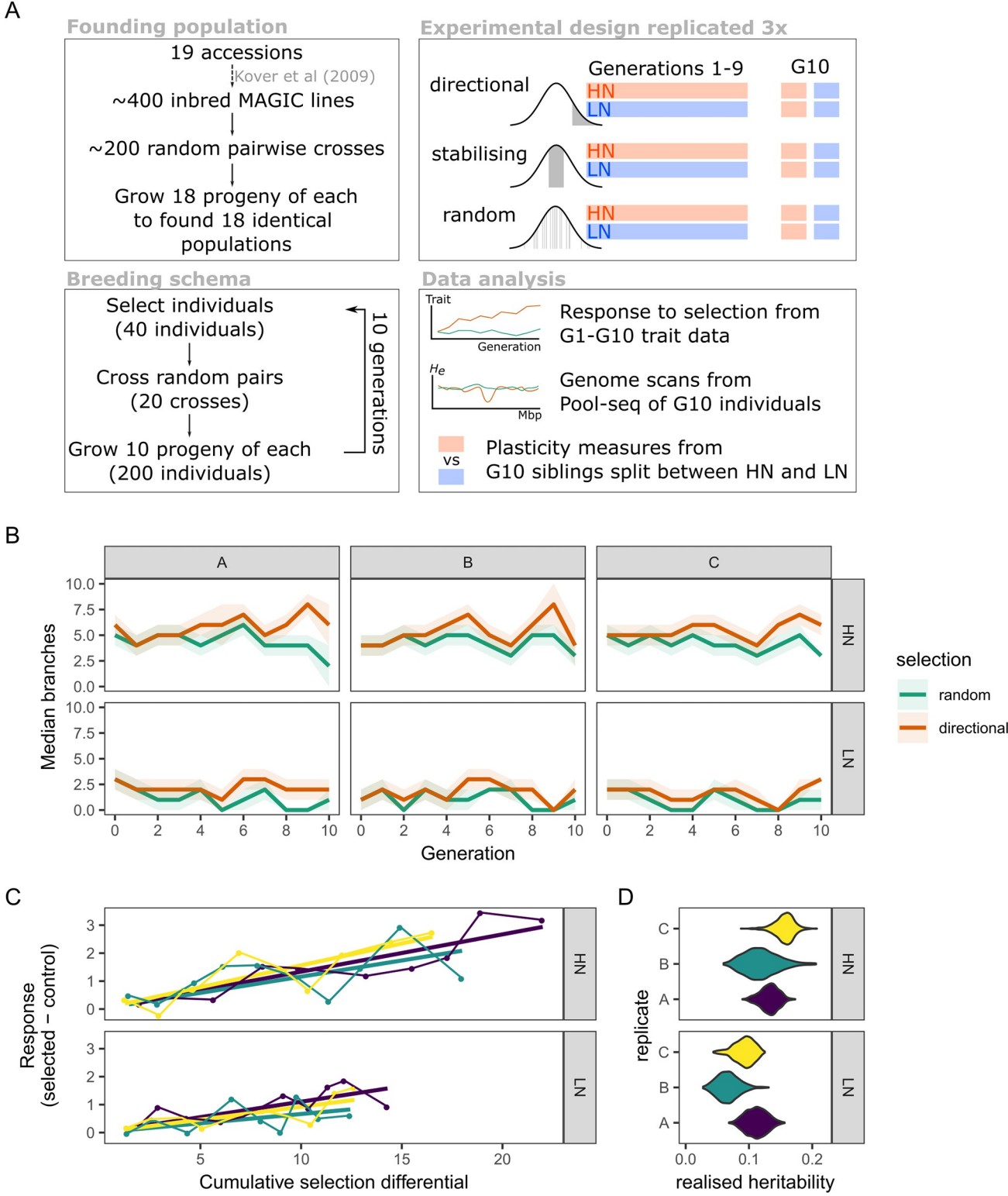

**Fig 1. Response to selection for increased shoot branching.** (A) Schematic of the selection experiment design. (B) Effect of selection for increased shoot branching across the generations in populations selected on high (HN) and low (LN) nitrate. The median number of branches in control (random) and selected (directional) populations is shown across the generations. The shaded areas show the median absolute deviation (a robust dispersion measure analogous to the standard deviation). There is an overlap between distributions, but the median of selected populations is consistently above that of the random controls. (C) Relationship between the cumulative selection differential imposed and the standardised response to selection. The selection

differential was calculated as the difference in mean branches between the selected individuals and that of the whole population. The standardized response to selection was calculated as the difference between the mean number of branches in the selected population minus that of the respective control population, which should control for some of the overall environmental variation (e.g. glasshouse conditions) between the generations [27]. The straight lines are the least-squares linear model fits, with an intercept at zero (i.e. no response to selection is expected when selection is zero). The colours distinguish the different replicate populations, as in (C). (D) Realised heritability estimates for each population, obtained from the slope of the fitted models shown in panel (B). The violin plots show the uncertainty of this estimate obtained by bootstrap resampling; in all cases estimates are above zero.

logistical constraints of performing crosses, in practice these numbers varied from our intended design (detailed in the methods), but despite this the genetic (and phenotypic) variance in all populations was similar at the beginning of the experiment. The census size of each population was kept at ~200 individuals, and in each generation 40 individuals (~20%) were chosen as parents for the next generation according to two selection regimes: directional selection and stabilising selection. Directional selection was performed by ranking plants by the number of elongated branches at the time when two fruits (siliques) were full and choosing the top 40 for reproduction. Stabilising selection was applied by choosing the 40 individuals with a branch number closest to the mean branch number in that generation. Thirdly, we included a control, where 40 individuals were chosen at random. For each population, the 40 chosen individuals were randomly paired and crossed and ~10 progeny from each cross were grown in the next generation, recreating the census size of 200 (i.e. each generation was composed of 20 full-sib families of size 10). The whole experiment was replicated 3 times, resulting in a total of 18 populations (2 nitrate treatments x 3 selection regimes x 3 replicates). Because these experiments were carried out under glasshouse conditions across a range of seasons, each replicate set (A, B, C) of selected/control populations was grown together, so comparisons between each selected population and their respective control take into account their shared environment.

We first assessed the effects of directional selection on the average number of shoot branches in these populations compared to control populations. Overall, there was a trend for the average number of branches to be above that of the equivalent random control population, although with some overlap between the distributions of the two populations (Fig 1B). This difference was more obvious on HN than on LN, but comparisons of the selected and control population distributions using a non-parametric effect size measure revealed this trend to be robust (S2 Fig). Despite some expected fluctuation, the probability that an individual from the selected population had a higher number of branches than one from the random population was >50%, with an increasing trend across the generations for both HN and LN in all 3 replicate populations (S2 Fig).

We next looked at the heritability of shoot branching in these experiments, using the breeder's equation (methods) [27]. As expected, the relationship between the response to selection and the selection differential applied had an upward trend in all replicate populations, allowing us to estimate heritability between ~6–11% on LN and ~11–13% on HN (Fig 1C and 1D). This is a measure of "realised heritability", which is likely an underestimate of the heritability in the source population, due to unaccounted random sampling effects [27]. These results show that the variation in shoot branching in the populations is heritable with enough genetic variation to respond to artificial selection, even in a relatively small number of generations.

We also assessed how the variability in shoot branching changed over time, in particular in populations under stabilising selection. There was no detectable response in the variation of branch numbers in these populations, which were indistinguishable from the control populations (S1 Appendix and S3 Fig). Incidentally, our analysis of shoot branching variation revealed a strong cyclical pattern that correlated with the time of year the populations were grown in the glasshouse (S1 Appendix and S4 and S5 Figs). This suggests interactions between shoot branching and multiple environmental variables such as nutrient availability and

seasonal conditions, further highlighting the complex regulatory networks involved in this developmental process.

## Selection for shoot branching affects its plasticity and flowering time

When selecting for a particular trait, other traits that are genetically correlated with it might also be selected, even if they are not the direct target of selection. We have previously identified several trait correlations in the founder MAGIC lines [23]. In particular, the response of shoot branching to N supply (shoot branching plasticity) correlates negatively with mean branch number on low N, but positively with mean branch number on High N. Thus non-plastic lines produce a moderate number of branches on both HN and LN, while highly plastic lines produce many branches on HN and very few branches on LN. Therefore, we hypothesised that selection for highly branched individuals on LN could result in lower plasticity genotypes, whereas the converse could be true for individuals selected for increased branching under HN conditions.

We tested this hypothesis by growing full siblings from the last generation of selection from each population under each of the two nitrate regimes (Fig 1A, methods). We then calculated the difference of the within-family average number of branches produced between HN and LN, to obtain an estimate of plasticity at the family level (Fig 2A). Our reason for estimating family-level plasticity is that measuring individual-level plasticity is only possible with inbred lines. For two of the three populations selected for increased branching on HN, the average plasticity increased by ~1.5 branches compared to random controls, with a weaker change in the other replicate (mean ± SE replicate A: 1.51 +/- 0.41, p = 0.007; replicate B: 0.84 +/- 0.43, p = 0.133; replicate C: 1.55 +/- 0.48, p = 0.008). Conversely, populations selected for high branching on LN had low within-family plasticity levels, similar to or lower than those of the control populations (replicate A: -0.64 +/- 0.44, p = 0.206; replicate B: 0.18 +/- 0.35, p = 0.617; replicate C: -1.18 +/- 0.36, p = 0.008). These results are compatible with our previous study of this trait's plasticity and suggest a genetic basis for these correlations, since the plasticity trait was co-selected with the branch number trait, depending on N supply.

We also investigated correlations with flowering time, since we had previously found that high branching on Low N correlates negatively with flowering time (S6A Fig) [23]. The correlations between branch number and flowering time in the selection populations were not consistent across the generations, with much fluctuation either above or below zero (S6B Fig). These fluctuations might be expected, given that flowering time is also sensitive to seasonal conditions, so correlations between the two traits might depend on how the variance of each trait is affected by seasonal cues. Given these correlation fluctuations, there was a low selection intensity for flowering time throughout the experiment (S6C Fig). Despite this, when comparing selected and control populations within each replicate (i.e. controlling for seasonal conditions), there was a trend consistent across replicates for slightly earlier flowering in populations selected for increased branching on LN (Figs 2B and S6D). This result is in agreement with the negative correlation between shoot branching and flowering time in the founder MAGIC lines mentioned above. It shows that early flowering is selected as a correlated effect of selection for increased branching under nitrate deficient conditions, suggesting a genetic correlation between the two traits.

## Genome scans reveal new QTL for shoot branching

We next investigated the genetic architecture of the selected trait by looking for signals of selection across the genome, based on allele frequency estimates obtained from whole-genome pooled sequencing (Pool-seq) [12,15]. Because our populations are descended from the

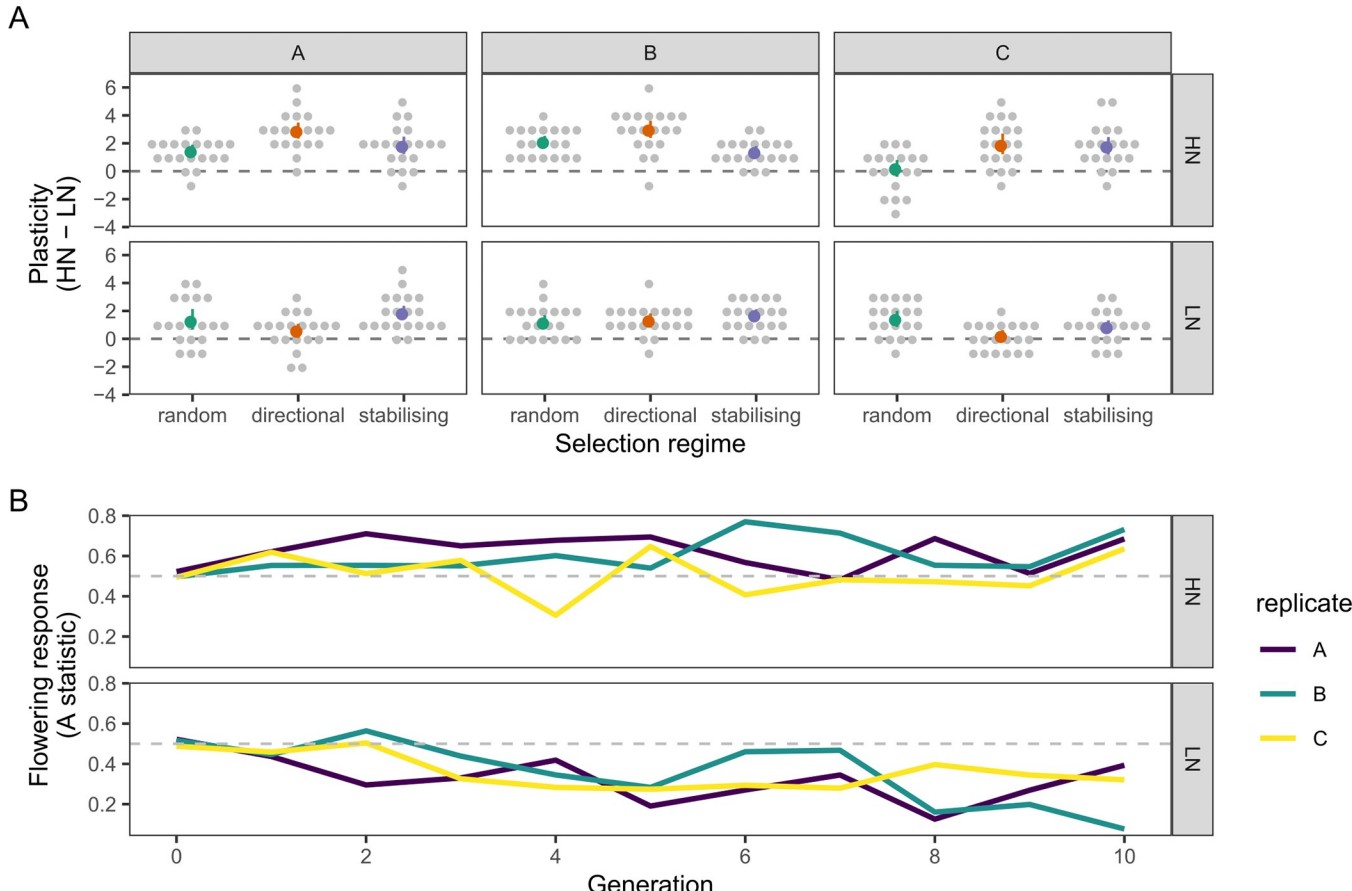

**Fig 2. Effect of selection for shoot branching on shoot branching plasticity and flowering time.** (A) Within-family response to nitrate at generation 10. Dotplots show the distribution of within-family responses to nitrate (plasticity), calculated as the difference in mean number of branches produced on HN minus the mean on LN within each family. The coloured points show the mean response across all families. Data are grouped by selection regime and replicate population. Plants were grown in the months of April (replicates A and B) and May (replicate C). (B) Effects of selection for increased shoot branching on the flowering time of individuals. The y-axis shows the Vargha-Delayney's A statistic [28], a robust non-parametric effect size measure expressing the probability that an individual from the selected population is later flowering compared to one from the control population. The grey dashed line represents A = 0.5, which is the null expected value of no effect of selection.

Arabidopsis MAGIC lines, which themselves are descended from 19 accessions with known genomes, we used a haplotype-based approach to infer the frequency of each of the 19 accession's genomes in 200Kb sliding windows across the genome [29]. As a means to summarise these 19 frequencies, we calculated genetic diversity in each window as the expected heterozygosity, $H_e$ (probability that two randomly sampled alleles are different). This measure of genetic diversity should be lower at a selected locus compared to the rest of the genome, assuming the selected allele rises to (near) fixation. This method also has the advantage of allowing us to assess the contributions of the individual parental haplotypes at a putative selected locus, which we would not be able to do if using SNP data alone.

On average, populations selected for increased branching had lower $H_e$ than either control populations or those under stabilising selection, and all populations had much lower $H_e$ than the founder population (Fig 3A). One hypothesis for this difference is related to differences in inbreeding between populations. Due to our experimental design, a certain amount of inbreeding is expected in all populations, as they go through a series of bottlenecks at each generation (where only ~20% of individuals, 40 plants, contribute to the next generation).

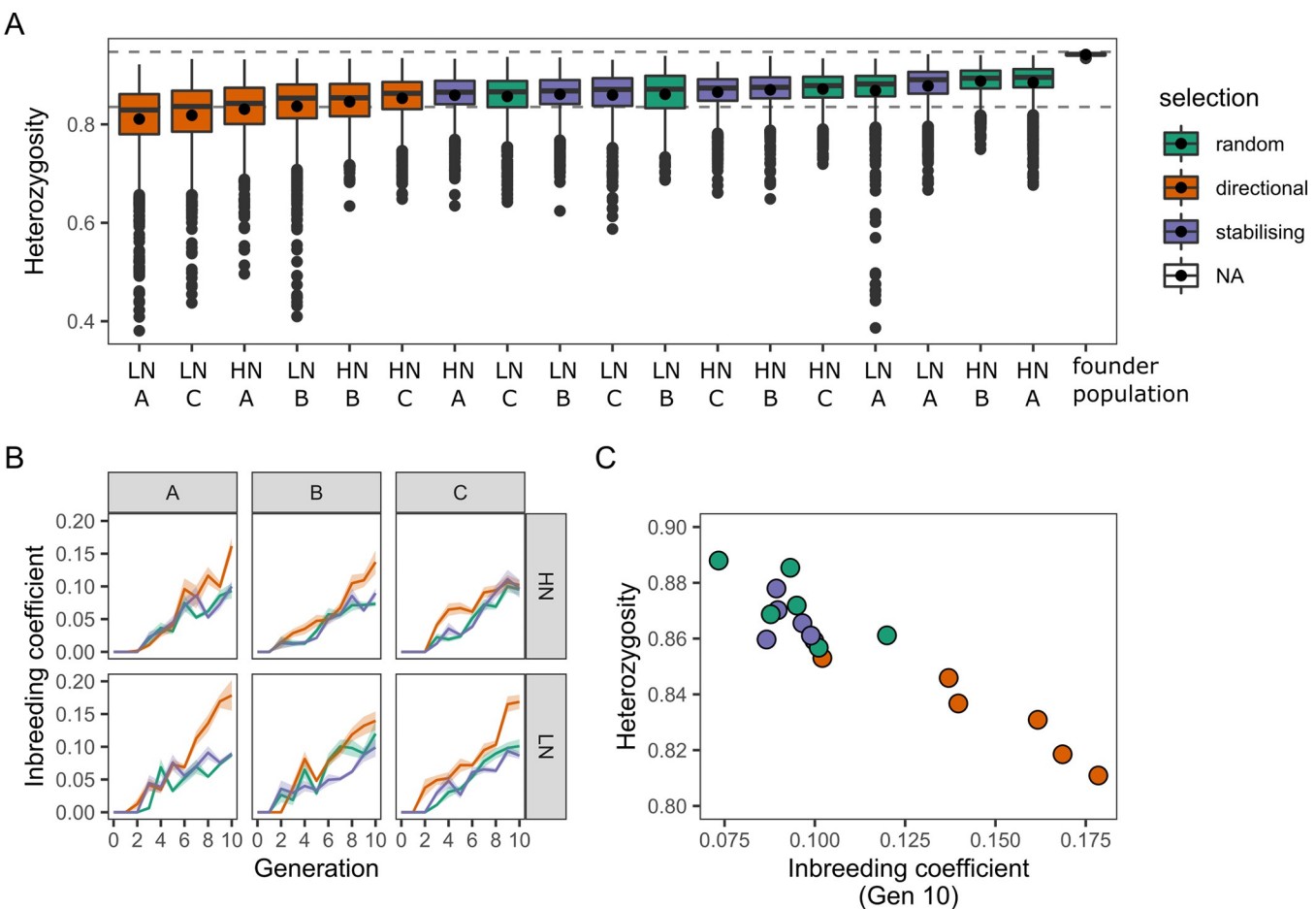

**Fig 3. Effects of selection on population inbreeding.** (A) Boxplots of expected heterozygosity ($H_e$) in all populations. Samples are ordered according to their median $H_e$ across all windows, with the respective nitrate treatment and replicate letter indicated along the x-axis. The mean $H_e$ is shown as a point inside each boxplot. The distribution of $H_e$ is also shown for the population of MAGIC lines used as founders for the selection populations. These have a tight distribution of $H_e$ close to the theoretical maximum shown with the upper dashed line (i.e. an equal frequency of the 19 Arabidopsis accession alleles present in these populations). The decay in heterozygosity in random populations was less than expected theoretically (lower dashed line): $H_t = H_0(1-1/2N)^t$ predicts that $H_{Gen10} \sim 0.83$ [30]. This may be due to associative overdominance, whereby recessive deleterious mutations are masked by remaining in a heterozygous state, leading to increased heterozygosity in linked regions of the genome [31,32]. (B) Change in the inbreeding coefficient across the generations for each population. There is an expected increase in all populations, however populations under directional selection reach higher inbreeding levels compared to control populations or those under stabilising selection. Data are shown for three replicate experiments (A, B, C) of populations selected on high (HN) and low (LN) nitrate. Lines are the mean inbreeding coefficient and shaded areas 2x the standard error (approximate 95% confidence interval). (C) Scatterplot showing relationship between mean $H_e$ and the mean inbreeding coefficient in the last generation of selection.

However, for a heritable trait under directional selection sibling selection might be expected to occur more frequently than by chance, resulting in an even higher inbreeding in those populations compared to controls. Indeed, based on the pedigree of each population, we saw an increase of the average inbreeding coefficient in all populations (Fig 3B). Crucially, populations under directional selection reached higher inbreeding coefficients in all replicates, expect one. As expected under our hypothesis, the average $H_e$ estimated from the sequencing data correlated well with the average inbreeding coefficient in the last generation of selection (corresponding to the generation of individuals that were pooled and sequenced) (Fig 3C). Populations selected for average branch number showed inbreeding trends comparable to random populations, again indicating that this selection regime had a negligible effect on these populations. We note that this analysis is based entirely on pedigree information within the

populations, and thus provides evidence for an effect of directional selection for shoot branching independent from, but consistent with, the phenotypic analysis presented earlier.

We next sought to identify the location of candidate selective sweeps in the genomes of these populations. From the $H_e$ distributions, it is clear that several of the selected populations have skewed distributions with low $H_e$ windows standing apart from the bulk of the data. These regions tended to cluster together, revealing several local $H_e$ decreases in populations under directional selection, compared to random controls (Fig 4). We considered these local decays in $H_e$ as candidate selective sweeps, whereby genetic diversity was locally reduced due to the selection of fewer haplotypes than would occur without selection. To define a putative selective sweep region, we used an empirical threshold for $H_e$ as the median of the 1% percentile of each of the 6 populations under directional selection. We defined the threshold based on the selected populations only because, as was mentioned above, they have on average lower $H_e$ across the genome to start with (Fig 3A, methods). Based on this threshold we identified several candidate selective sweeps across the six populations under directional selection (Fig 4). There was also one notable $H_e$ dip in one of the control populations, on chromosome 3 of replicate A on LN, suggestive of a false positive result. Such an extreme local decay of $H_e$ could be due to the effects of a local bottleneck impacting the diversity in this particular region (e.g. random selection of several individuals that shared a particular haplotype block in this region, leading to a rapid reduction of local diversity). Populations under stabilising selection had only one small region below our threshold, again compatible with no detectable effects of selection in these populations (S7 Fig). In conclusion, local drops in $H_e$ were more pronounced and frequent in populations selected for increased branching, suggesting that some of them at least should be real selective sweeps.

To understand better the genetic architecture underlying these putative sweeps, we looked at which accession haplotypes were selected in each region. We took advantage of knowing the 19 founder accessions that contributed to the populations to obtain their frequency spectra in each of these regions. At the putative selective sweeps, the frequency spectra were characteristic of hard sweeps, with one accession being substantially more common than all the others (Fig 5). Selection was not always for the same accession haplotype, with each of eight unlinked candidate sweeps involving a different accession (Fig 5B). This result suggests that a diversity of alleles contribute to increased shoot branching in these populations, with selection resulting in enrichment for different accession allele combinations in each population.

Surprisingly, there were few clear strong sweeps in the populations selected on HN, and in fact none in replicates B and C, despite robust evidence from the phenotype and pedigree data that they responded to directional selection. This may be partially explained by the fact that the background $H_e$ in these populations is affected by the increased inbreeding caused by selection. To investigate this, we estimated neutral $H_e$ distributions by simulating allele frequency fluctuations of 1000 loci "inherited" according to each population's pedigree. As expected, these neutral simulations revealed a negative correlation between $H_e$ and inbreeding (S8A and S8B Fig), similar to what was observed empirically. Using these simulations to define population-specific thresholds (adjusted for each population's inbreeding history) did not reveal new candidate sweeps for populations under directional selection (S8C Fig). However, there were 5 significant drops in $H_e$ in populations under stabilising selection (excluding two that fell on centromeric regions), suggesting these less stringent thresholds pick weaker signals in these populations (S8C Fig).

We then investigated whether multiple haplotypes were being selected, which may go unnoticed with our $H_e$ statistic as they would result in a "soft sweep" signature [33,34]. Our measure of $H_e$ is particularly low when only one of the 19 haplotypes is selected, but if multiple haplotypes are selected then it may not be sufficiently sensitive. We therefore used modified

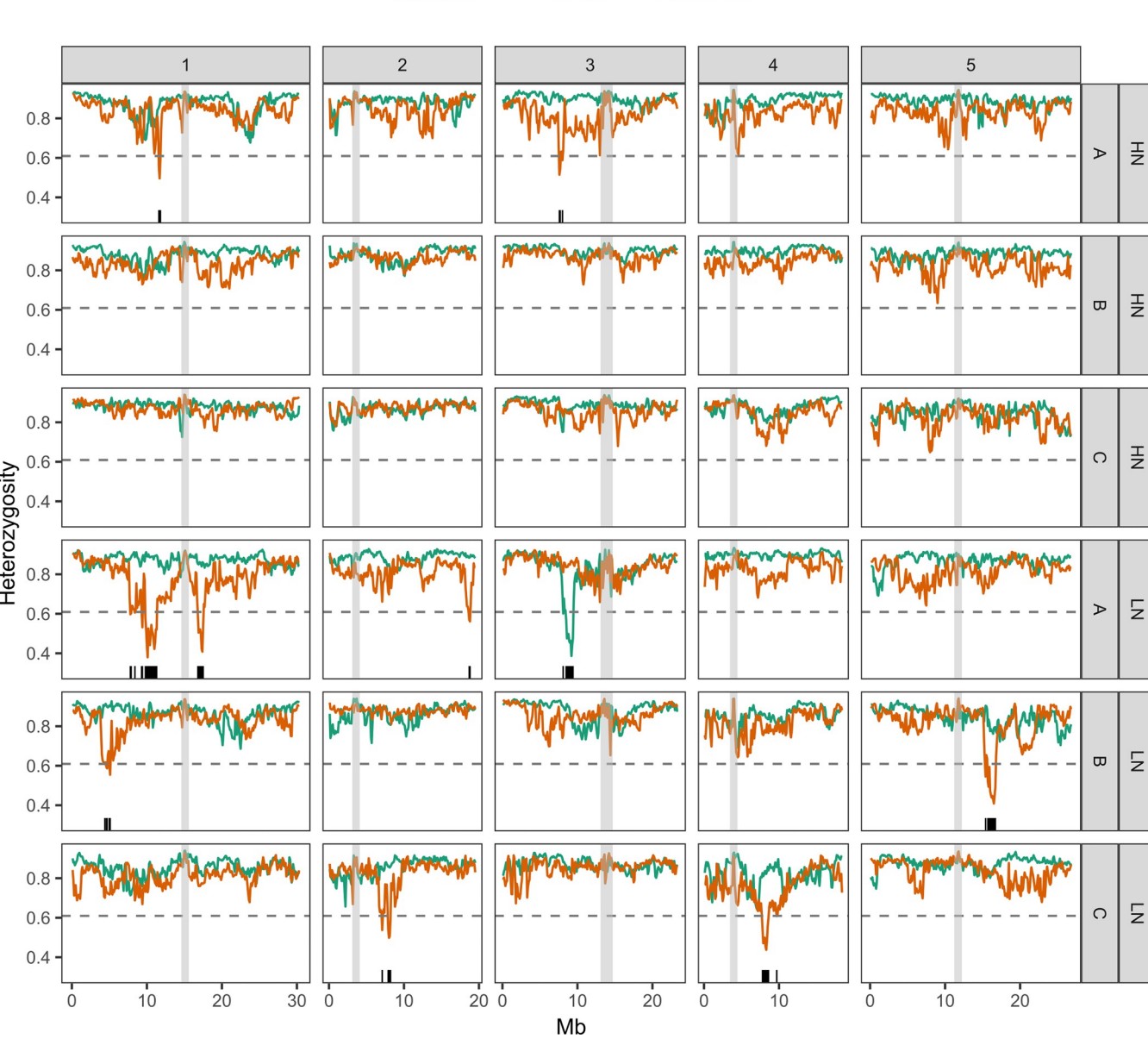

**Fig 4. Genomic scans of heterozygosity, H$_e$.** The plots show the profile of $H_e$ estimated from the haplotype frequencies of the 19 parental accessions inferred at 200Kb sliding windows (100Kb steps). Comparisons between control populations (green) and populations under directional selection (orange) are shown for each replicate under each nitrate treatment. The horizontal dashed line defines the significance threshold as the lower 1% percentile of the distribution in selected populations. The black lines along the x-axis highlight the intervals falling below this threshold. The shaded grey areas highlight 1Mb around each centromere, where mapping of sequencing reads is poor. Each horizontal panel shows a pair of selected and control populations for each nitrate treatment (HN, LN) and population replicate (A, B, C).

expected heterozygosity statistics where the frequencies of the two, three or four most common alleles are added together as if they were a single allele (we refer to these as $H_e12$, $H_e123$ and $H_e1234$, based on the homozygosity statistic of [35]). Most of the putative sweeps identified with these modified statistics coincided with the previous ones, albeit generally spanning larger blocks (S9A Fig). There were a few new putative sweeps identified in some of the populations, although these spanned small regions. In conclusion, this analysis did not reveal any new strong sweeps on HN populations, suggesting weak or incomplete selection for many loci

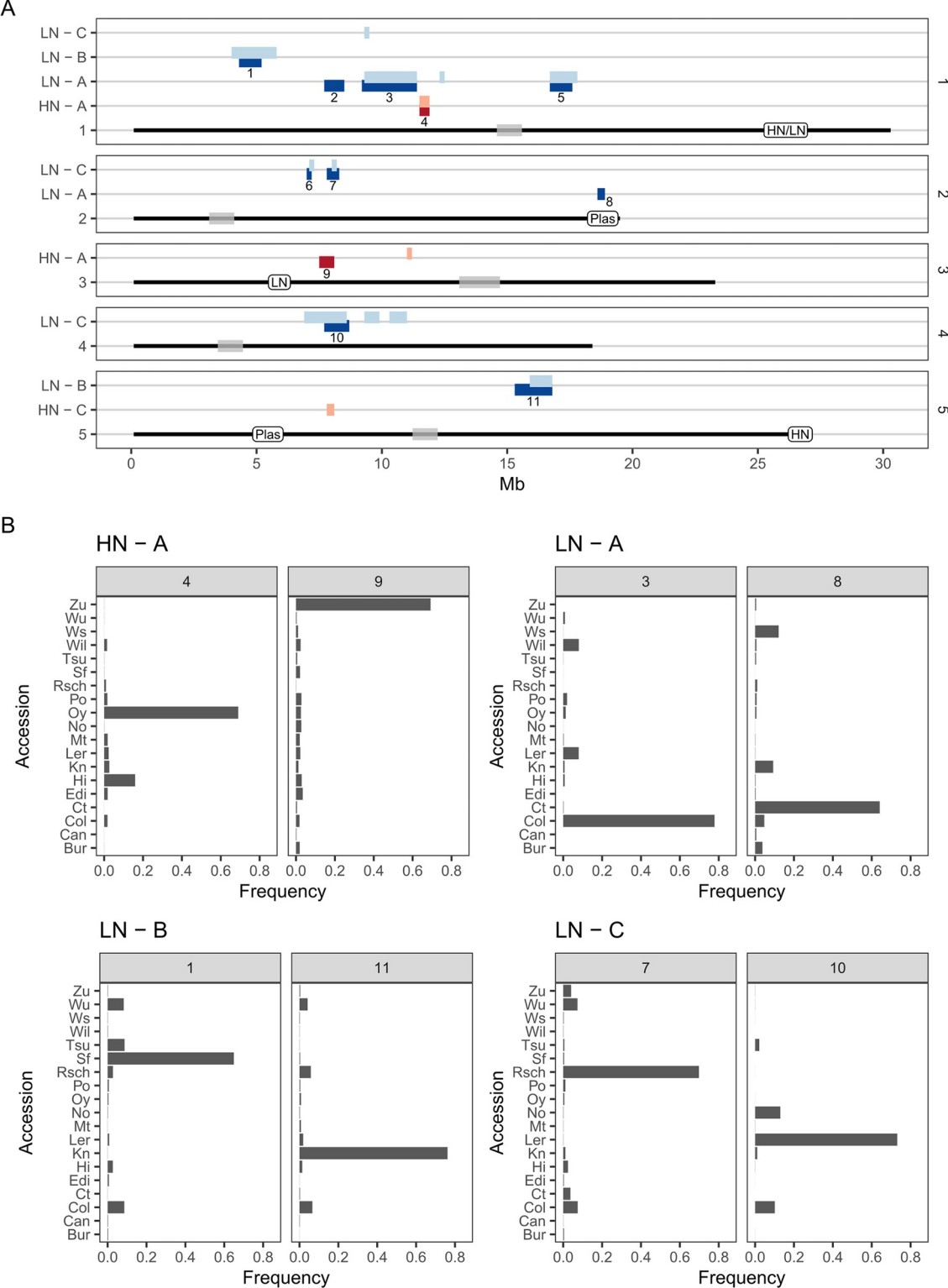

**Fig 5. Location and allele frequency spectra at candidate selective sweeps.** A) Schematic of the 5 Arabidopsis chromosomes showing intervals where heterozygosity was below the threshold of the 1% percentile in $H_e$. Intervals within 400Kb of each other were merged together (this corresponds to four consecutive sliding windows). Intervals are annotated with the population from which they derive: populations selected on low nitrate (LN, blue) and high nitrate (HN, red) for each of three replicates (A, B, C). The lighter coloured intervals correspond to candidate sweeps identified with the modified statistic $H_e12$, used to explore the occurrence of soft

sweeps. The white labels along the chromosome lines show the location of previously identified QTL for shoot branching in the MAGIC lines on low nitrate (LN), high nitrate (HN) and for the plasticity of shoot branching in response to nitrate (Plas) [23]. The shaded grey boxes indicate 1Mb centered on each centromeric region annotation. For reference, each interval is annotated with a sequential number and their locations summarised in S1 Table. B) Frequency spectra for the 19 accession haplotypes for representative candidate sweep blocks in each population. Each sub-panel is numbered as in A. Notice how alleles from different accessions were selected in each of the peaks.

of small effect in these populations compared to LN populations, where we find signatures more consistent with hard sweeps.

One of the $H_e12$ dips on Chr1 seemed coarsely to overlap between populations LN-A and LN-C (peak 3 in Fig 5A in light blue). We therefore investigated if the same accession haplotypes were selected in these regions. Indeed, the frequency spectra in these regions revealed some similarities, with accessions Col-0 and Wil-2 being the two most frequent haplotypes in both LN-A and LN-C populations (S9B Fig). In fact, Col-0 had a frequency >0.3 in all three LN populations. Even though for LN-B we did not find a heterozygosity dip in this region, the similarity of its allele frequency spectrum to the other two replicate populations may suggest an incomplete sweep. From this example, it seems that at least some of the outcomes of our artificial selection can be replicated, although the signal may be weak due to incomplete sweeps in some populations, which may be undetectable above the background. Another example of a coinciding weak signal occurred at ~8Mb on Chr5 of HN populations. In this case, the allele frequency spectra were not as repeatable (S9C Fig). The *Tsu* allele had the highest frequency in both HN-B and HN-C populations, but otherwise no other alleles consistently occurred at reasonable frequencies in these populations. This result raises the hypothesis that some loci contributing to shoot branching on HN involve multiple accession alleles (either due to shared causal variants or otherwise functionally equivalent alleles).

Finally, we complemented our analysis by performing a scan based on biallelic SNPs, i.e. ignoring the known 19 accession haplotypes (S10 Fig) [36]. We identified fewer peaks than with the haplotype-based approach, with two peaks being in common between approaches. However, two peaks were new: on Chr 2 population LN-A and on Chr 3 population LN-C. Looking at the inferred accession haplotype frequencies in those windows reveals weaker sweeps, with individual accession frequencies < 0.5 (S10B and S10C Fig). This may explain why these peaks were identified using a SNP approach, but not using a haplotype-focused approach. The two approaches are complementary, as the SNP-based approach suffers from the fact that, for most SNPs, the minor allele frequency is <10% (~50% of the 1.9M SNPs are private to a given accession) and thus their frequency often drops to zero due to drift.

Overall, our selection scans revealed a complex genetic architecture for shoot branching with candidate sweeps spread across the genome, few peaks shared between replicate populations, and several accession haplotypes selected. Furthermore, only one of these sweeps, on the right arm of Chr2, coincided with a QTL previously identified for shoot branching plasticity in the MAGIC lines, which were the founders of the selection populations (Fig 5A) [23]. Other QTL identified in that study did not clearly overlap with any of the sweeps. Our artificial selection approach thus revealed additional genetic complexity of shoot branching, with several putative loci identified, which had previously gone undetected in the QTL study.

## Simulations suggest distinct genetic architectures between nitrate conditions

Two main observations from our genome scans were: (1) on LN populations there is an inconsistent signal of putative sweeps across replicates and (2) on HN populations there are no clear sweeps, despite response to selection. We hypothesised that these could be explained by

multiple loci of small effect affecting the trait. To explore this hypothesis, we performed forward-time simulations of our experimental setup under different genetic architecture assumptions (see methods for simulation details). From each simulation outcome we calculated inbreeding, realised heritability (i.e. response to selection) and expected heterozygosity across the simulated genome. From an initial parameter scan, we defined parameter combinations that resulted in similar realised heritability to our empirical data (S11 Fig). The first observation from these simulations is that single-locus models are incompatible with the observed data, as there are only two outcomes: either the advantageous allele is lost early by drift and heritability is effectively zero; or the advantageous allele sweeps through resulting in a clear and consistent signal across replicate simulations.

Increasing the number of trait loci to 3 resulted in inconsistent signals across replicates. When there is a signal, it tends to be sharp and generally drops to near zero in individual simulations, but often a signal is found for only one or two trait loci, with some overlap between the distributions of heterozygosity at trait loci between random and selected populations. Such inconsistent signals are exacerbated when increasing the number of trait loci to 10 or 20 (S12A Fig). In this case the average genomic signals are weaker at trait loci, with a much higher overlap in the heterozygosity distributions between random and selected populations. The weaker signal in simulations with higher number of trait loci is a consequence of a lower allele effect, which we need to assume in order to remain within the realised heritability range compatible with our data (S11 Fig). This suggests that the inconsistent signal between replicates of our LN populations is to be expected for a trait that is controlled by ~10–20 loci (which is a range compatible with the 10 selective-sweep QTL we identified in LN populations, Fig 5).

As the number of advantageous loci increases further, the signal in individual simulations are less clear, and sometimes even indistinguishable from the background (e.g. at 30 loci with a phenotypic effect of 0.2 standard deviations S12B and S13 Figs). We could not put a higher bound on this number, as even assuming 500 advantageous loci of low effect (0.05 standard deviations) resulted in similarly weak signals, while still resulting in a positive response to selection (within the heritability bounds in the empirical data). Our simulations make several simplifying assumptions such as an equal spacing of advantageous loci across the genome, and an equal effect and starting frequency of the advantageous allele (starting at 1/19, i.e. a private advantageous mutation from a given accession). Relaxing these assumptions may change the exact numbers of predicted loci controlling this trait, but generally our simulations are compatible with ~2x or more loci involved in controlling shoot branching on HN compared to LN.

## Discussion

In this study we describe the results of a 10 generation selection experiment for shoot branching, designed to uncover features of the genetic architecture of this trait as well as its relationship to nitrate supply. We have found a clear response to directional selection, with a correlated response in plasticity and flowering time traits, supporting alternative "rapid exit" versus "extended foraging" strategies in response to N limitation [23]. Genomic scans of selection revealed that this trait is likely controlled by at least tens of loci, and its genetic architecture differs substantially between low and high nitrate conditions. Our simulations suggest that there could be ~2x or more loci controlling shoot branching on HN compared to LN, highlighting the complexity in the relationship between the environment and the developmental regulation of shoot branching.

### Effects of selection

We found convincing evidence that selection for high branching on both low and high N was effective, reflected both in increased mean branch numbers and in reduced genetic diversity

across the genome associated with inbreeding. The response was moderate, with realised heritabilities ranging from 6 to 13%, consistent with the relatively low broad-sense heritability previously estimated [23] and the relatively small number of generations of selection. The genetic tractability and relatively short life cycle of Arabidopsis makes its choice for this study appropriate, but even this small experiment required more than 3,500 manual crosses, making larger experiments with more statistical power challenging.

In contrast to the effects of directional selection, there was no apparent effect of stabilising selection for average branch numbers with respect to reduced variability, as assessed by the coefficient of variation (CV) in branching, or with respect to genomic diversity. These populations were mostly indistinguishable from the random selected controls. Directional selection for high branching on high N also had no effect on CV, but on low N it tended to reduce CV compared to control populations, however this was mostly due to the skewed trait distributions, rather than an actual selection on variance (S1 Appendix). Overall, this suggests a weak genetic basis for shoot branching variability, unlike what has been described for example in seed germination [37,38].

Selection for high branching on high N resulted in increased branching plasticity in response to N supply, whereas selection for high branching on Low N tended to reduce plasticity. That selection for high branching co-selects for plasticity is consistent with our previous findings that these traits correlate and that some loci appear to affect both branching and branching plasticity [23]. These results are also compatible with the so-called "Jinks-Connolly rule", which predicts that selection of a trait towards the population's cross-environment mean results in lower plasticity (i.e. selection for increased branching on LN), whereas selection away from that mean results in higher plasticity (i.e. selection for increased branching on HN) [39,40]. Several other empirical studies using artificial selection in plants have shown that the plasticity of a trait can be affected by changes in the evolved response to selection (reviewed in [22]). More specifically, a study using an ancestral population of the Arabidopsis MAGIC lines used here looked at the changes in flowering time plasticity, when flowering time was the target of directional selection under Spring or Winter conditions [41]. By selecting for early flowering individuals in either treatment, the authors found a correlated response in plasticity, whereby it increased for genotypes selected under Spring conditions but decreased for those under Winter conditions.

## Correlations between branching, season and flowering time

As well as correlated changes in plasticity, selection for high branching on low N resulted in slightly earlier flowering time, compatible with a negative correlation between shoot branching on low N and flowering time previously observed in the MAGIC lines [23]. The fact that shoot branching, its plasticity and flowering time remained coupled under artificial selection for branching suggests that regulation of shoot branching may be part of a broader phenology and/or life-history syndrome involving at least some common genetic control. Related to this, we observed a clear pattern in the distributional properties of shoot branching across generations that strongly correlated with the months in which plants were being grown (independently from the selection regime). The regularity of the pattern observed raises questions about the relationship between branching, flowering, nitrate availability and seasonal conditions. Both shoot branching and transition to flowering are regulated by the interplay between external and internal cues, so it is perhaps not surprising that there are some correlations between these traits, and that they are co-selected in our experiment, since they may respond to the same cues via the same regulatory networks. We note that flowering time has also been implicated in axillary meristem formation [42].

## Nitrate-specific genetic architecture for shoot branching

Many heritable traits have a complex genetic architecture, which may involve a combination of multiple loci (polygeny) each with small effect, rare alleles, allelic heterogeneity, locus-by-locus interactions (epistasis), and gene-by-environment interactions [6,9]. These properties make it statistically challenging to dissect the genetic architecture of complex traits based on association mapping (QTL and/or GWAS) and may explain the discrepancy between heritability estimated from phenotypic data and the fraction of it that is explained by mapped loci [4,43,44]. Indeed, previous attempts to map the genetic basis for shoot branching and its plasticity in response to nitrate supply met with limited success [23]. Some QTL were identified in the MAGIC population, providing evidence for both shared and independent genetic contributions to branch numbers, branching plasticity and flowering time. However, only a fraction of the heritability estimated for shoot branching in that study was explained by the QTL identified. The current study highlights why this must have been the case: the genome scans and simulations allowed us to estimate that tens of loci of low effect likely underpin this trait. In fact, there was remarkably little overlap between the candidate selective-sweep-QTL identified and our previous QTL study on the MAGIC lines (only one overlap at interval #8 in Fig 5A). Thus, previous mapping experiments using a few hundred individuals were likely statistically underpowered, in particular in multi-parent populations such as the MAGIC lines [45]. Our estimate for the number of QTL is a lower bound, as there is low power to detect very small effects. In addition, multiple causative loci may in fact occur on the same selected haplotype block, and we have no resolution to distinguish this from linked selection on neutral loci [46,47].

The well-defined population ancestry of our selection lines allowed us to carry out haplotype-based analysis of genetic diversity across the genome [29]. By analysing both the genomic patterns of expected heterozygosity and the underlying haplotype frequency spectrum of each, we found an inconsistent pattern between different populations, including between replicates grown under the same nitrate conditions. Across eight broadly defined selective-sweep-QTL we found selection for eight different accession haplotypes (Fig 5B). This varied genetic outcome suggests that there is genetic redundancy for shoot branching, i.e. that different allelic combinations may achieve the same overall phenotype [48,49].

There are many examples in Drosophila where experimentally imposed pressures result in correlated trait responses across replicates, but with distinct genetic outcomes (reviewed in [50]). From earlier studies of ethanol tolerance [51,52] to more recent studies of desiccation tolerance and adaptation to hot environments, amongst others [16,18,53]. While the population of origin is critical in the genetic response to selection [54], even populations with similar founders can show parallel evolution for a trait through selection for different loci. Simulations show that this pattern can be explained by genetic redundancy, whereby populations converge toward similar phenotypes with different underlying allelic combinations across many loci, as long as the founder population has enough genetic diversity and the trait has a relatively complex genetic architecture [16,55,56]. This is compatible with our own data and simulations.

There was also a discrepancy between the genomic signals on LN and HN populations. Most candidate loci were identified in LN populations, with only a few weaker signals in HN populations. Our simulations indicate that this result is expected for a trait controlled by tens of loci, but most notably that the number of loci is predicted to be higher for HN populations than LN populations. This is compatible with the idea that alleles at loci controlling shoot branching are conditionally advantageous/neutral, and could explain the high genotype-by-environment interaction observed for this trait [23,57–59].

It remains to be clarified how the genotypic redundancy described here relates to the underlying developmental pathways controlling shoot branching, in particular with respect to the

interaction with nitrate availability. Is the genotypic redundancy associated with functional redundancy (e.g. loci may be functionally redundant if they affect rate-limiting steps of the same biochemical pathway), or are different pathways involved in regulating this trait [60]? A study of inbred lines of Arabidopsis suggests that functional redundancy is not common in this system [61]. However, this study focused primarily on t-DNA lines, which are likely to carry knockout mutations and thus it remains unclear whether natural alleles affecting different developmental programs show functional redundancy in this system. Further dissection of the causal loci affecting shoot branching variation may help address this question in the future.

Another possible factor contributing to disparate selective-sweep-QTL alleles in each population is epistasis, which makes the effect of any one allele conditional on genetic background [14,62]. Epistatic interactions are often ignored in QTL/GWAS studies due to the statistical challenges associated with detecting them, although epistasis may well be of importance both for domesticated species as well as in natural conditions [63,64]. However, epistasis may be of modest effect if selection is relatively weak in relation to drift, which is likely to be the case in our experiments [62,65]. While our data are consistent with these hypotheses, further genetic analysis will be needed to test them more robustly. For example inbred lines carrying different accession alleles at the candidate selective-sweep-QTL could be used to determine whether they have additive or non-additive background-dependent effects on shoot branching.

The question remains about whether the trait complexity described here is of relevance in wild populations, as that would require, for example, performing field experiments using ecotypes adapted to different environments, assessing the frequencies of relevant alleles in those populations and where they might be adaptive [66–68]. The same is true for the issue of epistasis just mentioned: while this may be of relevance in breeding experiments, it may bear no significance in the evolution of wild populations, in particular if the allele frequencies at the interacting loci make it unlikely for an individual to carry the right combination of alleles [69].

The strong selective sweeps we detected were characterised by enrichment for a single haplotype, out of the 19 accession haplotypes present in our population. Given the population size and number of generations in our experiment, we lacked the statistical power to detect weaker sweeps [55,70,71]. We attempted to do so by using modified $H_e$ statistics used to detect "soft sweeps" [33–35] as well as SNP-based scans using biallelic loci [59,72]. However, our results were only modest, with only 4 new candidate loci identified in this way, 3 of them in LN populations and only one in HN populations (further emphasising the difference between nitrate conditions). In these cases, multiple accession alleles seemed to be selected for, which could either be because these accessions share common causal SNP variants or because they have functionally equivalent alleles, despite their different sequences. Distinguishing between these two would require finer investigation of candidate genes, as for example has been done for regulators of flowering time and seed germination [73,74].

In summary, the disparate patterns of selection across replicate populations and the allele frequency spectra underlying each selective-sweep-QTL, suggest that shoot branching is underpinned by at least tens of loci of small effect, and this is supported by our simulations. As mentioned earlier, shoot branching regulation likely interacts with other regulatory pathways, both developmental (such as hormone signalling/perception and flowering) and in the response to the environment (such as seasonality-related signals). Such regulatory networks are likely to be highly interconnected, and thus small-effect alleles across a wide range of loci, even those seemingly unrelated to the trait under study, may in fact have an impact on a trait such as shoot branching [75]. It will be interesting to investigate the overlap between our candidate QTL and their positioning in such regulatory networks, for example by exploring existing large-scale transcriptome networks [76] or by exploring co-expression patterns more specifically across a range of mutants in "core" shoot branching regulators (such as *MAX*, *SMAXL* and *BRC* genes) [24,77]. Furthermore, given the difference between LN and HN

populations we observe, it may be fruitful to explore the transcriptomic changes in young buds from plastic and non-plastic genotypes growing under different nitrate conditions (i.e. if and how much the trait plasticity correlates with their transcriptome plasticity) [78,79].

### Candidate loci

While our study provides new insights about the underlying genetic architecture of this complex trait, further work would be desirable to validate the candidate selective-sweep-QTL we identify and fine-map the causative loci. The mapping intervals we define are large and therefore contain many genes each, any of which could be causative to the underlying increase in shoot branching.

There is one locus at the end of Chr2 that overlaps between the QTL mapping results we presented previously and a selective-sweep-QTL we present here. Neighbouring this locus (~240Kbp away from the lowest $H_e$ window) is the *MAX3* gene, which encodes an enzyme involved in strigolactone biosynthesis [80]. Strigolactone is a major regulator of shoot branching and loss of strigolactone production results in a high branching phenotype on low N [25]. Within this region we also find the *SOC1* gene, involved in flowering time regulation, and whose expression and effect on flowering has been shown to be nitrate-sensitive [81]. Given the observed correlation between flowering and shoot branching on LN, this gene should be a good candidate for future detailed studies, alongside *MAX3*.

We have also found a new mutation in the *MAX2* gene in a LN population under directional selection (S2 Appendix). *MAX2* is required for the response to strigolactone, and its loss of function results in a highly branched phenotype, regardless of N supply [82]. In a population with limited genetic diversity, such as those used in experimental/artificial selection experiments, once standing genetic variation is exhausted as a source for phenotypic change, only new mutations can contribute to the increase of the trait under selection [12]. However, with such a low number of generations as in our study, it seems unlikely that new mutations significantly contributed to the selective responses observed. Consistent with this, no unusual genomic signal around *MAX2* was detected, suggesting that this new mutation had yet to contribute significantly to the selected trait.

### Conclusions

In summary, by combining artificial selection with haplotype-based genome scans, we were able to provide a more detailed picture of the genetics underlying shoot branching variation in Arabidopsis. Despite its relatively low heritability, this trait responded consistently to directional selection across several replicate populations, and our analysis and simulations suggest a lower bound of tens of loci underpinning this trait.

Selection for high branching resulted in correlated effects on shoot branching plasticity and flowering time, which we have previously shown to be relevant for fitness under different nutrient regimes [23]. This suggests that the regulatory network that controls and links these traits can be tuned through selection for high branching in multiple different ways to give similar phenotypic outcomes (earlier flowering and low plasticity on LN; late flowering and high plasticity on HN). Our data also support that a smaller number of loci contribute to the ability to maintain high branch numbers on low N vs high N, which has implications for future work on the mechanistic basis for this trait's developmental plasticity.

### Materials and methods

#### Plant material and growth conditions

As the starting material for our populations we used 400 Arabidopsis MAGIC lines, which are derived from the random inter-crossing of 19 accessions [26]. Growth conditions, low and

high nitrate treatments, and trait measurements were as described in [23] and further details can be found there. Briefly, plants were grown in a low-nutrient soil mixture (50/50 sand and terragreen), which allowed finer control over the nutrient provision to the plants. Once a week plants were fed with *Arabidopsis thaliana* salts (ATS) solution, containing either 9mM (high N treatment, HN) or 1.8 mM $NO_3^-$ (low N treatment, LN), and otherwise watered with tap water as needed. Flowering time was scored as the number of days from germination to the day at which the flowering bud was visible in the center of the rosette. Branch number was counted as the total number of secondary shoots >1cm in length, from both rosette and cauline leaves together. These traits were measured when plants had formed two full siliques. Plants were grown under glasshouse conditions at the University of York between 2009 and 2013.

### Selective breeding experiment

To initiate each population, the 400 founder MAGIC lines were randomly paired and crossed to produce 200 F1 families. Because the MAGIC lines are inbred, the siblings of each of these F1 families were genetically identical. We thus assigned one F1 individual from each family to initiate 18 genetically similar populations, each assigned to a particular combination of nitrate growth conditions (LN and HN), selection regime (directional, stabilising, random), and replicate (3 replicates named A, B, C). In practice, several lines did not flower on time for these crosses, and the initial number of MAGIC lines that founded each population varied between 308 and 342 (median 324), producing between 154 and 171 (median 162) F1 individuals in each population. Because the MAGIC lines originate from 19 accessions, whose allele frequencies are equally represented in the population, these slight differences in numbers between populations should not impact on our analysis, as the sample size (>300) is well above the number of possible alleles (19 accessions). For example, assuming an allele at 1/19 frequency, the 95% intervals of its expected frequency derived from a binomial distribution using a sample size of 308 (our smallest founder population) and 342 (our largest) are [0.029, 0.078] and [0.032, 0.084], respectively.

Once at least 80% of the plants in a population were scored for branching, 40 individuals were chosen (according to the selection regime) and randomly paired to produce 20 crosses. From each of these crosses, 10 progeny were grown, resulting in a census size of 200 at each generation. In this way, we kept track of the entire pedigree of each population.

We set the 80% scoring cut-off due to the logistic constraint of having all individuals flowering simultaneously in order to inter-cross them manually. This may have imposed some additional selection for example on slow growing or very late flowering individuals (e.g. those needing vernalisation). However, this should have affected all populations equally, including the random selection controls.

The selection differential imposed (the difference between the average number of branches of selected individuals and the population average) was ~1–3 branches and was slightly higher in populations grown on HN compared to LN (S1A Fig). This difference was related to differences in trait variance in the two environments, with populations with higher variation having a higher selection differential (S1B Fig). Indeed, by scaling the selection differential by the standard deviation of the trait in each population at each generation, we observed a relatively constant selection intensity of ~1 standard deviation across the generations (S1C Fig). There was one notable exception, in replicate C grown on HN, where it is apparent that selection was relaxed at generation 3. This was due to a randomisation error during the choice of individuals for crossing, which resulted in an effectively random selection of individuals in that generation.

## DNA extraction and sequencing

In the last generation of selection we collected seed from all 200 self-fertilzed individuals of each of the 18 populations. We then pooled together 5 seeds from each (total of 1000 seeds per population) and grew them in liquid media for 10 days. We used 50ml liquid ATS media, shaken at 40 RPM in a tissue culture room with 16 hour light and a temperature of 22/18°C during day/night cycles. After 10 days, all seedlings were harvested and ground together for DNA extraction. In this way, each of the original 200 individuals should have contributed a similar amount of tissue to the final pool.

DNA extraction was carried out using a CTAB method. Seedlings were drained from the ATS media and dried on tissue paper, then flash-frozen in liquid nitrogen and ground to a fine powder using a pestle and mortar. 5 ml of CTAB buffer (100 mM Tris pH 8.0, 1.4 M NaCl, 20 mM EDTA, 2% CTAB, 0.2% β-mercaptoethanol) was added to ~1g of tissue and incubated at 65°C for 20 minutes. Samples were then left to cool to room temperature and 2.5ml $CHCl_3$ was added and vortex-mixed. Samples were centrifuged at 13,000 x g for 5 min, and the aqueous phase transferred to a new tube. 2ml of isopropanol was added and gently mixed to form a DNA precipitate, which was pelleted by centrifugation at 13,000 x g for 5 min. The pellet was washed with 70% ethanol and then resuspended in 100 ml TE buffer (with added RNase) at 4°C overnight. DNA was re-precipitated by adding 10 ml 100% ethanol with 400 µl of 3 M NaOAc (pH 7.0) and centrifuging at 13,000 x g for 5 min. After leaving the pellet to air dry, the DNA was finally resuspended in 50 µl TE buffer.

Sequencing libraries were prepared using the TruSeq DNA Sample Prep Kit (Illumina) following the manufacturer's protocol. Libraries were sequenced by the Genomics Core Facility at the Cancer Research UK Cambridge Institute on a HiSeq 2500 using a TruSeq SBS Kit-HS sequencing kit, to produce 100bp paired-end reads.

We aimed for >100x average depth of coverage for each sample. However, 6 of our 18 samples did not meet this requirement (HN-directional-B, HN-stabilising-A, HN-stabilising-B, LN-directional-A, LN-random-A and LN-stabilising-B). Therefore, we re-extracted DNA and prepared new libraries for resequencing. In this case, seedlings were grown on compost (rather than liquid media) and a size hole punch of leaf tissue was used as material for DNA, which was extracted using DNeasy Plant Mini Kit (Qiagen) following the manufacturer's protocol. The sequencing libraries were prepared using the TruSeq DNA PCR-Free Low Sample kit (Illumina) and sequencing was done on a NextSeq 500/550 machine using a NextSeq 500/550 High Output Kit v2 (Illumina) sequencing kit, to produce 150bp paired-end reads. We checked for sequencing batch effects, but the results were identical using either the first (lower depth) or the latter (high depth) data. The analysis presented in the paper uses the higher depth data.

## Response to selection

To assess the response to selection of each trait, and obtain estimates of realised heritability, we used the breeder's equation: $h^2 = R/S$, where $R$ is the mean response to selection and $S$ is the selection differential [27]. $R$ was calculated as the difference in the mean trait value of the selected population minus the mean in the respective control population for each generation. The selection differential, $S$, was calculated as the difference between the mean of those individuals selected for breeding minus the mean of the whole population in that generation. When plotting cumulative $S$ against $R$, the slope of the regression line fitting that relationship gives an estimate of $h^2$ [27]. We estimated uncertainty in this estimate by bootstrap-resampling the data for each population across the 10 generations (1000 bootstrap samples).

Since the full pedigree of each population was known, we also calculated inbreeding coefficients for each individual, using the *pedigree* R package [83].

## Family-level branching plasticity

To assess the impact of branching selection on the plasticity of this trait, we split the final generation of each population into two groups, one grown on LN and another on HN (regardless of the nitrate regime under which they had been selected). The splitting was done at the family level, i.e. the 10 siblings originating from each cross were split into two groups of 5. In this way, we were able to estimate average plastic responses to nitrate, adjusted for the genetic relatedness of individuals belonging to the same family. This was done by fitting a multilevel model with family-level random intercepts (branch number on LN) and slopes (response to HN). As fixed effects we included the effect of selection regime, nitrate treatment and their interaction. The model was fitted using the R/lme4 package [84]. Informally, the model fitted to the data was (following lme4 syntax convention):

branches ~ nitrate + selection + nitrate:selection + (nitrate|family)

The contrast we were interested in was the selection:nitrate interaction term between selected and random (control) populations (i.e. the difference in plasticity between these populations). The results of this contrast were obtained using the R/emmeans package [85]. The model was fitted separately to each selection population replicate (A, B and C).

## Processing of sequence data

All bioinformatics analysis scripts are available online at https://github.com/tavareshugo/publication_Tavares2023_Nselection, so we give only a summary of the pipeline used. Raw reads were quality checked using FastQC (0.11.3) [86] and filtered to trim adapters and remove low quality reads. Some of the samples contained bimodal GC distributions, which was attributed to contamination of the liquid cultures used to grow the seedlings used for DNA extraction. These led to a reduced sequencing depth of coverage after mapping to the Arabidopsis genome, as those reads clearly belonged to a different organism. Alignment of reads was against the *Arabidopsis thaliana* TAIR10 genome assembly (Ensembl release 37) using *bwa mem* (0.7.12) [87]. Duplicates were removed using the *markduplicates* tool from *Picard* (2.21) [88], followed by realignment around indels using *RealignerTargetCreator* and *IndelRealigner* tools from *GATK* (3.8) [89]. *samtools* (1.10) [90] was used for basic manipulation of the alignment files and obtaining mapping statistics.

To help with reproducibility the analysis of these sequence data (including haplotype frequency estimates detailed in the next section) is provided as a *Snakemake* workflow [91].

## Estimating haplotype frequencies

Haplotype frequencies were inferred using *harp* [29]. This software requires a set of known SNPs of the genomes expected to be segregating in the pool. We used published variants from the 19 Arabidopsis accessions used to breed the MAGIC lines [92], downloaded from http://mtweb.cs.ucl.ac.uk/mus/www/19genomes/variants.tables/ (accessed Apr 2023). These were filtered to retain only biallelic sites, resulting in 1,993,761 SNPs. Haplotype frequencies were estimated in window sizes of 10Kbp, 100Kbp, 200Kbp and 500Kbp, sliding in steps half of the respective size. The patterns described in the results were similar across this range of window sizes, and data shown in the paper refers to windows 200Kbp wide only.

The choice of the mapping software used in the pipeline described above was carefully considered, due to potential mapping biases—and thus distorted haplotype frequency estimates—when mapping pooled sequence data [93]. This is particularly relevant in our case, since one of the accessions in our population is Col-0, the same as the reference genome. Taking the ~1.9M known SNPs from the 19 accessions, we first noted that a disproportionate number of non-reference (different from Col-0) SNP alleles had a frequency of zero in our pools, even if their

predicted starting frequency in the founding population of MAGIC lines was ~50%. This likely indicates a mapping bias against non-reference reads, leading to an inflated frequency of Col-0 alleles. We tested if this bias differed depending on the mapping software and reference genome. We tested the mapping algorithms of *bowtie2* (2.2.5) [94], *stampy* (1.0.29) [95] and *bwa mem* (0.7.12) [87]. For each of these we further used as a reference genome either the standard sequence, or a version where known variants across these accessions were masked with an ambiguous nucleotide 'N'. *bwa mem* had substantially less "allele loss" compared to *bowtie2*, and was comparable to *stampy* (although this software was substantially slower). Masking known variants in the genome had no impact on the results from *bwa mem*. This result agrees with previous analysis on *Drosophila*, where *bwa mem* was one of the top-performing tools [93].

Because we didn't have a ground truth for what the frequency in our pools should be, we performed a further check using artificial pools. We used publicly available genome data (EBI ENA Study PRJEB2457) for 16 of the MAGIC founder accessions that had a similar type of sequence data (Col-0, Hi-0 and Mt-0 did not have equivalent data, so we excluded them to avoid sequence quality biases). We used these data to create artificial sequence pools where each of the 16 accessions was represented equally. These pooled reads were aligned to the genome using *bwa mem* and we estimated haplotype frequency of each accession across the genome using *harp*. In this analysis *harp* was given SNP information for all 19 accessions, and so we expected the 3 accessions that were excluded from our artificial pools to have a frequency of zero, whereas all others should have an equal frequency of 6.25%. The result revealed that most accessions did not deviate from this expectation. The highest bias was against No-0 with a difference of ~1% below expectation. Despite not being included in our artificial pools, Col-0, Hi-0 and Mt-0 were still estimated to have a median frequency of ~3%, ~2% and ~1.6% respectively. We re-ran the pipeline excluding Col-0 as one of the possible parents given to *harp* for estimation of the frequencies. This did not improve the overall frequency across the other haplotypes, suggesting that any bias induced by reads being identical or diverged from the reference genome does not affect any particular haplotype, but is rather spread across all of them. The biases were substantially larger in peri-centromeric regions, which we highlight in our figures and refrain from making strong conclusions about any patterns found there.

In conclusion, using *bwa mem* for mapping provided the least biased estimates in our data. Any remaining bias seems to affect all haplotypes equally and its magnitude is smaller than the signal we detect in our selective-sweep scans.

## Calculating expected heterozygosity

Expected heterozygosity was calculated for each genomic window as the probability of sampling two distinct alleles by chance:

$$H_e = 1 - \sum_{i=1}^{19} p_i^2$$

where $p_i$ is the frequency of the *i*th most common allele. In this case, $p_i$ corresponds to the frequency of each of the 19 accessions estimated by *harp*. Modified heterozygosity statistics were calculated based on the *H12* statistic of:

$$H_e = 1 - (p_1 + p_2)^2 + \sum_{i=3}^{19} p_i^2$$

Essentially, this statistic considers the two most common alleles as if they were a single allele, allowing $H_e$ to get very low values if, together, those two alleles were much more common than the rest. This can be generalised to any number of alleles:

$$H_e = 1 - \left(\sum_{i=1}^{x} p_i\right)^2 + \sum_{j=x+1}^{19} p_j^2$$

where $x$ is the number of most common alleles that get added up as if they were a single allele. We varied $x$ from 1 (the standard statistic) to 4.

## Thresholds for identifying candidate selective sweeps

We used two strategies to define thresholds for identifying candidate selective sweeps: an empirical threshold and a simulation-based threshold. The empirical threshold was defined as the 1% percentile of the distribution of expected heterozygosity observed in populations under directional selection. Our reasoning was based on the fact that these populations have, on average, lower heterozygosity (due to higher inbreeding) and a threshold based on their $H_e$ distributions resulted in a low false discovery rate when used in control (random selection) populations.

In a second approach, we calculated population-specific thresholds by simulating null $H_e$ distributions based on the specific pedigree of each population. We simulated 1000 independent loci, each with 19 alleles at equal starting frequency (an assumption justified based on the empirical haplotype frequencies in MAGIC lines, which were randomly intercrossed [26]).

## Population simulations

We performed forward-time simulations of our whole selection experiment under different genetic architecture assumptions, and made similar estimates from these as for the empirical data (heritability due to response to selection, expected heterozygosity across the genome and inbreeding). We assumed five chromosomes, each with a number of loci corresponding to the number of windows from our genome scans (303, 195, 233, 184, 268), which should also emulate the chromosome sizes of Arabidopsis. Each locus had 19 possible allele states (equivalent to the 19 accessions in our experiment). We made a simplified assumption of constant recombination rates across the genome, with an average of 2 recombination events per chromosome (equivalent to sampling from a Poisson distribution with rate 2) [96]. Each population was initialised with 200 diploid individuals, with their genotypes at each locus sampled from the 19 possible alleles with equal chance. This initial genotype sampling should emulate the slight deviations from a fixed 1/19 frequency, which is also expected in our empirical data.

After determining each individual's genotype, they were assigned a "trait" value, which was sampled from a normal distribution with mean zero and unit standard deviation. The effect of an advantageous allele was then set to increase the mean of this distribution. This parameterisation allows us to interpret the allele effect as the number of standard deviations by which the allele increases the trait.

At the end of each generation the 40 individuals with the highest value for this "trait" were picked to form the next generation. Similar to our experimental setup, we did this by randomly pairing those individuals and generating a progeny of size 10 from each. Note that by setting every allele to have an effect of zero, this sampling is effectively equivalent to random selection of individuals (i.e. our control populations).

From this basic simulation setup, we then varied three main parameters: number of "trait loci" (loci harboring at least one allele which increased the trait value); number of

"advantageous alleles" at those loci; and the "allele effect" of those alleles (i.e. their effect on the trait average). Another parameter that could have been changed is the location of the "trait loci". To simplify our simulations, we always placed these in different chromosomes (if less than or equal to 5) and then at equal distances within each chromosome (if greater than 5).

The simulations were coded using the python package *simuPop* [97] and the code is provided at https://github.com/tavareshugo/publication_Tavares2023_Nselection.

### Genotyping the new *MAX2* mutation

To identify the generation when the new mutation in *MAX2* occurred, progeny of our manual crosses were grown from seed, to trace the mutation back through the generations. We note that the genotyped individuals were siblings of the actual individuals that were selected (since we did not have material for every individual in the experiment). Genotyping was done by Sanger-sequencing a fragment of *MAX2* containing the new mutation. The fragment was amplified by PCR using the following primers:

- forward 3'-CTTCCCTCTTTCTTCTTCTAAA

- reverse 3'-TAAAATGGACCAGTTTCTGAAG

    The following sequencing primers were then used:

- reverse 3'-ACGGCGGAGAGAGTCTC

- reverse 3'-CGAATGAATACTTGCACCTC

- forward 3'-GTATCACCAAATCTTGCCCTA

- forward 3'-GGAGGTAGAAGGAAGAGTG.

    Variants within this fragment were used to identify in which of the 19 accessions' allele the new mutation had occurred in. Because Col-0, Can-0 and Kn-0 had identical *MAX2* sequences, neighbouring diagnostic SNPs were used to distinguish them. SNPs diagnostic for Can-0 and Kn-0 were located 6315bp and 7294bp downstream of *MAX2*'s start codon, respectively. A single genomic fragment containing these diagnostic SNPs was amplified using the following primers:

- Forward 3'—TTTCCAATATGTGACTTTGC

- Reverse 3'—GTTGTGTTGCTTTAAGATGA

    And sequenced using the following primers:

- Forward 3'—CTCACACATAATACTTTAACGT (for Can-0 SNP)

- Reverse 3'—ACGTTAAAGTATTATGTGTGAG (for Kn-0 SNP)

### Data analysis

Data analysis and visualisation were carried out using the statistical software *R* version 3.4.1 [98]. The meta-package tidyverse [99] was used for data manipulation and visualisation. Other specific analysis or statistical tests are described in the relevant sections above or in figure legends. All analysis scripts are providedat https://github.com/tavareshugo/publication_Tavares2023_Nselection.

## Supporting information

**S1 Fig. Selective breeding for increased shoot branching.** (A) Selection differential across the generations, calculated as the difference between the mean number of branches of selected

individuals and the overall mean number of branches of the population in each respective generation. For each replicate, the selection differential is shown for selected and control populations, which for the latter fluctuate around zero as expected, shown by the dashed line. Populations were selected under two different nitrate supply regimes, low (LN) and high (HN). There were independent populations, A-C, for each selection regime and N environment, making a total of six populations. (B) Relationship between variation in shoot branch number (measured as standard deviation) and the imposed selection differential. The selection differential is strongly correlated to trait variation. Coloured points distinguish the 3 replicate populations. (C) The selection intensity (calculated as the selection differential shown in B divided by the trait standard deviation). Selection intensity is relatively constant across the generations and between selection environments. There is an exceptional dip in selection intensity in generation 3 of replicate C on HN, which was due to a randomisation error assigning individuals to crosses (see methods).
(TIF)

**S2 Fig. Response to selection for increased shoot branching.** The response to selection measured using the Vargha-Delayney's A statistic [28], a robust non-parametric effect size measure expressing the probability that an individual from the selected population has a higher number of branches than one from the control population. Replicate populations are coloured as indicated in the key (populations from the same replicate were grown simultaneously in the glasshouse). The grey dashed line represents A = 0.5, which is the null expected value of no effect of selection.
(TIF)

**S3 Fig. Selection on shoot branching variation.** See S1 Appendix for more details about this figure. (A) Standard deviation (SD) differential for shoot branches, calculated as the difference between the standard deviation in shoot branch number for the selected individuals minus that of the overall population. Data are shown for the directional selection populations, for the stabilising selection populations and for the random controls. In both kinds of selection there is a negative SD selection differential, which is as expected: in the case of stabilising selection we sampled from the centre of the distribution; under directional selection, we sampled from the upper tail of the distribution. In both cases, there is an imposed differential selection on the dispersion of shoot branching. As expected, there was no SD selection differential in control populations. (B) Changes in the median number of branches in the random control populations, and populations selected for average branch numbers across the generations. The shaded areas show the median absolute deviation (a robust dispersion measure analogous to the standard deviation). Stabilising selection had no effect on the median number of branches, which might have been expected if random selection had affected the mean of the trait differently from the imposed stabilising selection. In all panels data are shown for three replicate experiments (A, B, C) and for populations grown on high (HN) and low (LN) nitrate.
(TIF)

**S4 Fig. Effects of selection and season on shoot branching variability.** See S1 Appendix for more details about this figure. (A) Effect of selection on shoot branching variability, measured as the coefficient of variation (CV = standard deviation divided by the mean). The response to selection was calculated as the difference in CV between populations subjected to selection (directional or stabilising) and the respective controls in each generation. (B) Coefficient of phenotypic variation in shoot branch number in each population across the generations. (C) Correlation between the sowing month and shoot branching CV. Each point is the CV calculated for one generation and population. The trend lines are a smooth fit by local weighted

regression.
(TIF)

**S5 Fig. Characteristics of shoot branching distributions.** See S1 Appendix for more details about this figure. (A) Examples of distributions of shoot branch numbers from HN and LN populations with different CV values. The distributions shown are for selection generation 9. The numbers within each panel are the CV of the respective distributions. The sowing months are indicated for each replicate. (B) Distribution of shoot branch number by sowing month shown as histograms coloured by the CV. On low N, distributions are highly skewed (leading to a high CV), with a mode of zero branches in the Autumn/Winter months. Populations selected for high branching on low N have a heavier upper tail than the unselected control or those selected for average branch numbers. Replicates and generations were combined in these plots, since we were interested in exploring the marginal effect of scoring season on shoot branching.
(TIF)

**S6 Fig. Relationship between shoot branching and other traits.** (A) Scatterplot of number of shoot branches versus days from sowing to flowering for 357 of the MAGIC lines that founded the selection populations. Data are the median of each trait calculated from 4–8 replicates of each MAGIC line (median n = 7). The rank-based Spearman's correlation (rho) is shown in each panel with 95% bootstrap confidence intervals in brackets. Data are from [23]. (B) Spearman rank-order correlation between shoot branching and days to flowering across the generations in the selected populations. The shaded areas show the 95% bootstrap confidence intervals (1000 bootstrap samples). The dashed line at zero indicates no correlation. (C) Selection intensity for days to flowering across the 10 generations. (D) Changes in median days to flowering across the generations for populations selected for increased branching and random control populations. The shaded areas show the median absolute deviation (a robust dispersion measure analogous to the standard deviation).
(TIF)

**S7 Fig. Genomic scans of heterozygosity, $H_e$.** Identical to Fig 4 in the main text, but showing the result for populations under stabilising selection.
(TIF)

**S8 Fig. Defining heterozygosity thresholds adjusted for population-specific inbreeding.** A) Neutral heterozygosity distributions for 1000 simulated loci obtained by using random "inheritance" on each population's pedigree. Populations are ordered by their median heterozygosity. The ranking is similar to the empirical data shown in Fig 4A. B) Correlation between simulated heterozygosity and inbreeding coefficient of each population, showing that these simulations recapitulate what is observed empirically (compare with Fig 4C). C) Empirical genome scans of heterozygosity for each population, scaled by their respective simulated neutral distributions. The scaled values represent the number of standard deviations the observed heterozygosity is away from the mean of the simulations. The arrowheads highlight the windows falling below -4 standard deviations (dashed line). Data show estimates in 200Kb sliding windows with a 100Kb step. See methods for further details on the simulations.
(TIF)

**S9 Fig. Genomic scan with modified $H_e$ statistics to identify soft sweeps.** A) Putative selective sweep blocks identified with modified $H_e$ statistics [35]. These are calculated as previously described, but adding the frequency of the 2, 3 or 4 most common alleles as if they were a single allele ($H_e12$, $H_e123$ and $H_e1234$, respectively along the y-axis). For example, if there was a

sweep where 4 alleles were equally selected to the detriment of the other 15 accession alleles, adding their frequencies should create a very common "meta-allele", resulting in low expected heterozygosity. Similarly to Fig 4 in the main text, the blocks were identified based on falling below 1% of the distribution across selected populations. Blocks within 400Kb of each other were merged together. Shaded grey regions show 1Mb around the annotated centromeric regions. B) Allele frequency spectra at ~10cM on Chr1, which has common sweeps between LN—A and LN—C populations, highlighted with an arrow head in panel A, and corresponding to the region of peak #3 in Fig 5A of the main text. Population LN—B is also shown, as it shows some similarities in its allele frequency spectrum to the other two LN populations. C) Similar to panel B, but for a shared weak signal on Chr5 of HN populations (indicated by an arrow head in panel A).
(TIF)

**S10 Fig. SNP-based genomic scans.** These scans are similar to the heterozygosity-based analysis (Figs 4 and S9), but focusing on biallelic SNP changes (rather than accession-based frequencies). We show the absolute frequency change—|AFC|—between generation 10 (estimated from the Pool-seq data [36]) and generation 1 (inferred based on the SNP allele carried by each accession). A) Mean |AFC| summarised across 200kb windows (step size of 100kb). Outlier windows were defined as those falling above the 99% percentile of the distribution of |AFC| in each selection regime. These are indicated with an arrow head. Some of these peaks had also been identified with the heterozygosity-based analysis, but two are new: on Chr 2 population LN-B; on Chr 3 population LN-C. B) and C) show the estimated frequencies of each accessions' allele for these two new peaks on all three LN populations, with chromosome and approximate position shown on each plot's title.
(TIF)

**S11 Fig. Distribution of realised heritability from simulations.** The varying parameters in our simulations were: number of loci affecting the trait (colours); the effect of each of those alleles on the trait (y-axis), which was scaled as the number of standard deviations of increase above the trait's mean; and the number of accession alleles at a locus that increase the average trait value (panels), which is equivalent to the starting frequency of the advantageous allele in the founder population (all accession alleles start at an equal frequency of 1/19 to capture the random breeding of the MAGIC lines [26]). The grey shaded area highlights the interval [0.05, 0.2], which is broadly compatible with our empirical estimates of realised heritability (Fig 1D).
(TIF)

**S12 Fig. Genomic patterns of expected heterozygosity in simulated populations.** A) and B) show results for different parameter combinations as labelled in the plot titles. Panels i) show average genomic pattern of heterozygosity across 100 replicate simulations under random or directional selection. The line shows the median value and the shaded regions the 10% and 90% percentiles across replicate simulations. The upper triangles show the location of trait loci. Panels ii) show the distribution of heterozygosity at one of the trait loci (the first locus on Chr1, but the distributions are similar for the other trait loci). Panels iii) show the percentage of simulations with a certain number of trait loci falling below the 1-percentile of the genome-wide distribution of heterozygosity. Note that in A) most replicate simulations have at least 1 trait locus below this threshold and heterozygosity distributions are largely non-overlapping. Whereas in B) there is a higher overlap in heterozygosity distributions and in the number of simulations with no trait loci below 1-percentile.
(TIF)

**S13 Fig. Genome scans from individual simulation replicates for a trait controlled by 30 equally-distributed loci.** These are individual replicates from the simulation average shown in S12B Fig. Notice how in some of the simulations the signal around trait loci is hard to distinguish from the background (most strikingly for sim82, but to an extent also sim83 and sim84). The signals are also very inconsistent across replicates (i.e. different trait loci are selected for each time).
(TIF)

**S14 Fig. New max2 mutant allele arose during selection.** See S2 Appendix for more details about this figure. A) Representative photos of Arabidopsis lines Col-0 (wild-type), *max2-2* mutant, and an inbred line (ID 4210) that was produced from single seed descent of individuals carrying a new *MAX2* mutation that arose during the selection experiment. Plants were grown on compost for 8 weeks. The photo on the left highlights the increase in branch number and shorter height of the mutants; the photo on the right highlights the difference in leaf shape. B) DNA haplotype network of known *MAX2* alleles based on publicly available data. The haplotypes are for the 19 accessions that founded the MAGIC lines. Each node of the network represents a haplotype (size proportional to the number of accession/mutant alleles), and edges connect closest haplotypes (based on the Hamming distance between each allele). The size of the edges is proportional to the number of polymorphisms separating each allele (indicated on the edge, except for alleles with only 1 polymorphic difference). The node containing the standard wild-type accession allele, Col-0, is in blue, with two EMS-derived mutant alleles in dark blue (*max2-1* and *max2-2*). The new allele is shown in green.
(TIF)

**S1 Table. Location of candidate selective sweeps in populations under directional selection.** See Fig 5A for further details.
(PDF)

**S1 Appendix. Extended analysis of the effects of selection and environment on shoot branching variability.**
(PDF)

**S2 Appendix. Extended analysis of the new MAX2 mutation arising during selection.**
(PDF)

## Acknowledgments

For their excellent technical support, we would like to thank: Julie Affleck, Julie Mercer, Ann Barker, Ekaterina Kozhevnikova, Mariya Budarina, Petra Stirnberg, Lisa Williamson, Rebecca Butler, Rachel Borrows and the Horticultural Technicians Teams at the University of York and the Sainsbury Laboratory Cambridge University. We would also like to thank Stephanie Smith for her comments on early versions of the manuscript. Finally, we thank Paula Kover for providing us with the Arabidopsis MAGIC lines and for discussions in the early phases of this project.

## Author Contributions

**Conceptualization:** Hugo Tavares, Ottoline Leyser.

**Data curation:** Hugo Tavares, Anne Readshaw.

**Formal analysis:** Hugo Tavares, Raj K. Pasam, Liron Shenhav, Elizabeth Forsyth.

**Funding acquisition:** Ottoline Leyser.

**Investigation:** Hugo Tavares, Anne Readshaw, Urszula Kania, Maaike de Jong, Raj K. Pasam, Hayley McCulloch, Sally Ward.

**Methodology:** Hugo Tavares, Anne Readshaw, Sally Ward, Ottoline Leyser.

**Project administration:** Anne Readshaw, Sally Ward, Ottoline Leyser.

**Software:** Hugo Tavares.

**Supervision:** Hugo Tavares, Anne Readshaw, Ottoline Leyser.

**Visualization:** Hugo Tavares.

**Writing – original draft:** Hugo Tavares.

**Writing – review & editing:** Hugo Tavares, Anne Readshaw, Urszula Kania, Maaike de Jong, Raj K. Pasam, Hayley McCulloch, Sally Ward, Ottoline Leyser.

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
