## [Decision Letter · Decision Letter 0]

17 Oct 2022

Dear Ottoline,

Claudia Köhler asked me to "guest edit" this paper — apologies for the slow decision, which was partly for personal reasons, partly because this is a complex paper, and it was difficult to find reviewers. In the end, this paper has been reviewed by four people, and I have also read it. The reviewers had quite different opinions, including on whether this paper should be accepted or not, largely dependent on their background. They did, however, agree that it was an impressive study and that the results are solid, so based on this we are willing to publish a major revision, even though the results are arguably not surprising.

What needs work is the analysis and interpretation. The paper is currently framed in terms of the unfortunate "missing heritability" debate. I say "unfortunate", because even the people who helped coin this phrase have long since admitted that this simply reflects ignorance of basic statistics and quantitative genetics: you just don't have the power to pick up anything but the largest effects. There is no reason for researchers outside human genetics to be drawn into this discourse. Especially not when you have a design like yours, where power is extremely limited and linkage disequilibrium extensive — a very different setting from enormous human GWAS studies. You keep talking about the architecture being "polygenic", but what do you actually mean by that? Can you put bounds on how many loci are involved, and can you estimate their effect sizes? The latest human height GWAS identified 12k putatively causative loci — perhaps there are 12k loci affecting roots, but you obviously cannot say anything about this. My point here is that "polygenic" ends up being yet-another buzzword, and it would be more helpful to use your simulation results to try to delimit what could actually be going on (to the extent this is possible, and to the extent you want to make claims about it).

Likewise you keep mentioning allelic heterogeneity, but what does this actually mean? How large are the haplotypes that are actually segregating in this population, and that are being selected? It seems to me that unless the number of causative sites/loci is very small, you will almost certainly have heterogeneity among haplotypes.

In general, there is a tendency to try to apply various concepts from population genetics where they do not quite make sense. For example, as noted by Reviewer 1, using heterozygosity to look for "selective sweeps" is odd given that you will have no power to detect anything except the rise of singleton haplotypes. Furthermore, you discuss the possibility of genetic drift without mentioning the classical result that heterozygosity should start at 1 - 1/19, and decrease by a factor (1 - 1/80)^10 — which it doesn't appear to do, suggesting that maybe there is some associative over-dominance in the population.

It is obviously none of my business to tell you how to interpret your results, as long as all conclusions are backed up by data, but the whole thing feels overwrought, and risks being misleading. Far better to focus the analyses what conclusions may actually be drawn from the (impressive) data using a simpler and more bottom-up way, then put it into context in the Discussion.

Magnus

[Begin form letter, which I don't have time edit in light of the above, please interpret accordingly]

Thank you very much for submitting your Research Article entitled 'Artificial selection reveals complex genetic architecture of shoot branching and its response to nitrate supply in Arabidopsis' to PLOS Genetics.

The manuscript was fully evaluated at the editorial level and by independent peer reviewers. The reviewers appreciated the attention to an important problem, but raised some substantial concerns about the current manuscript. Based on the reviews, we will not be able to accept this version of the manuscript, but we would be willing to review a much-revised version. We cannot, of course, promise publication at that time.

If you decide to revise the manuscript for further consideration at PLOS Genetics, please aim to resubmit within the next 60 days, unless it will take extra time to address the concerns of the reviewers, in which case we would appreciate an expected resubmission date by email to plosgenetics@plos.org.

We are sorry that we cannot be more positive about your manuscript at this stage. Please do not hesitate to contact us if you have any concerns or questions.

Yours sincerely,

Magnus Nordborg

Guest Editor

PLOS Genetics

Claudia Köhler

Section Editor

PLOS Genetics

Reviewer's Responses to Questions

**Comments to the Authors:**

Reviewer #1: Tavares et al described an artificial selection approach coupled with Pool-seq aiming to identify the genetic components contributing to shoot branching under various nitrogen conditions. The application of artificial selection in plants is rather novel and seems to be a heroic effort. The discovery of a new MAX allele during the artificial selection is interesting.

My major concern of the manuscript is that although the approach is novel, the uncovered genetic insights underlying either shoot ranching or nitrogen response seem to be limited. The independent replicates do not confirm each other to prioritize genomic regions of interest, and the discussion of underlying gene loci in the sweeps was rather limited. This could be caused by minor contribution of many loci, however, it also could be caused by the limitation of experimental system such as the small population size and short number of generation.

Overall the manuscript read a bit descriptive and technical, which could benefit from editing and streamlining to focus on the central biological question (branching and nitrogen)

About the nitrogen effect, since at least part of the motivation was to study the underlying mechanism of shoot branching plasticity in response to N changes, I was wondering what difference it would make if the selection was done by picking the highly N-responsive plants (ranked by difference between the branching in low N and high N)? At least the author should discuss the logic of the selection design.

Minor:

1) Authors should describe low N and high N conditions in result session 1

2) Figure 6 was chopped off

Reviewer #2: Tavares et al present results from an artificial selection experiment on shoot branching in Arabidopsis. They monitor 18 populations, characterizing the phenotypic responses to selection and then applying pooled sequencing to the generation 10 descendants. At the trait level, responses to selection were not large, but owing to the high level of power in their design, the authors were able to establish a genetic basis to variation (realized heritabilities > 0.0). This is not the sort of thing that PLoS genetics typically publishes, but I appreciated Tavares et al’s thorough workup on the direct and correlated responses to selection.

The genome sequencing of the evolved populations is the second part of the study. Here, I am confused by their approach. Tavares et al focus on He, the amount haplotype variation within each part of the genome. It is true that classic selective sweeps will cause localized reductions in He. However, this seems a very indirect way to identify selected loci in an experiment that was maintained for 10 generation. What researchers typically look for in these kinds of studies is whether particular alleles change in the same direction within treatments but not across treatments. The ‘high allele’ for branching at a QTL should consistently increase in the up-selected populations but not control or stabilizing treatments. If 4 of the original 19 lines carried the high allele, the selection populations might show a change from 20% to 50% for these haplotypes (collectively). If this happened consistently across selection replicates, the locus would likely be highly significant by any of the standard evolve-and-resequence tests (reviewed recently by Schlötterer and colleagues), but would show minimal signal in He. These tests can be applied SNP to SNP or on haplotypes. Given the fairly weak inferences obtained from He, I could not see any reason why these more standard analyses would not work with this dataset.

Reviewer #3: The general aim of the paper is to understand the genetic architecture of shoot branching in Arabidopsis thaliana. The authors have already carried out classical QTL and molecular studies on the trait and they here complement those by carrying out artificial selection experiments. More precisely they created populations by crossing inbred lines (MAGIC lines) and applied different selection schemes: directional selection and stabilising selection over 10 generations under both Low and High nitrate as well as a control population w/o artificial selection. The study included three replicates. They also sequenced the lines at the start and after 10 generations of selection (G10) allowing them to look at signature of selection at the genome level. They focused on selective sweeps although those , given the type of trait studied, do not appear very likely from the start. Overall the paper is a really nice piece of (large!) work, is well written and the analyses were carried out carefully. Perhaps the paper contains too many angles for its own good (e.g the part on the max2 mutant) but I would be inclined to keep them as they also make the paper more complete.

The results are not very surprising: response to directional selection differs under different environments, and shoot branching is likely highly polygenic and redundant (many combinations of loci can give the same result), something that has been observed repeatedly in recent human genetic studies, in particular in a series of recent studies by Jonathan Pritchard's group (e.g. Sinnott-Armstrong, N., Naqvi, S., Rivas, M. & Pritchard, J. K. GWAS of three molecular traits highlights core genes and pathways alongside a highly polygenic background. Elife 10, e58615 (2021) and reference therein). The omnigenic model proposed by Pritchard could be relevant here, at least for discussing the results. In general the authors could discuss a bit more the strengths and limitations of their study. For example, one general issue of this type of study that the authors might also want to discuss is the difficulty to draw evolutionary inferences from the type of study they have carried out. In a natural population the distribution of allele frequencies is likely highly different from that in the artificial population they created by crossing MAGIC lines. Since the effect of an allele substitution depends on allele frequencies, there is a risk that the genetic architecture detected in artificial populations and under artificial environmental conditions only has a vague resemblance to that observed under natural conditions. This departure between artificial and natural site frequency spectra can, for instance, explain why epistatic variance matters in population created by crossing lines fixed for alternative alleles while it does not seem to be significant in other populations (Hill, W. G., Goddard, M. E. & Visscher, P. M. Data and theory point to mainly additive genetic variance for complex traits. Plos Genetics 4, e1000008 (2008)). As the authors rightly point out in the introduction "In addition, allelic diversity at each locus, population structure, and the effect of environment on the traits under study all complicate the analysis" but it also strongly impact the meaning of the conclusion that can be drawn. So the authors could increase a bit the scope of the discussion and in particular place their results in a broader context.

Minor comments:

Figure S1 is very helpful and could be integrated in the main text.

Line 422. This will not be correct if there is balancing selection and it should be pointed that there will be a large variation among loci/haplotypes due to drift.

Reviewer #4: Tavares & Readshaw et al. investigated the genetic basis of branch number in Arabidopsis. The authors’ goal was to understand the genetic architecture of shoot architecture and plasticity in shoot architecture through a very ambitious select and sequence strategy. To accomplish this, the authors used an Arabidopsis MAGIC population with 19 founders. They then selected this population for 10 generations under nitrogen limited and high levels of nitrogen. They conducted both stabilizing and directional selection. As expected, response to selection was only seen in the directional selection populations. Sequencing the populations at the end of selection allowed the authors to establish the genetic architecture that was important for changes in shoot architecture under each selection regime. Overall, there were 18 populations that were selected upon, which were broken into sets of three replicates for each type of selection regime. The authors found that a different genetic architecture was under selection in low nitrogen selections in contrast with the high nitrogen selections. Low nitrogen suppresses branching. From genome-wide data, the authors identified genomic regions that were likely under selection over the generations of the project. These results contrasted between selection in high and low nitrogen. To better understand those results, the authors conducted computational simulations. Finally, the authors discovered a new mutation in one of their experimental populations that affected shoot architecture. Using complementation studies, they showed that the new mutation was in the MAX2 gene.

I am overall very impressed with the study design and the amount of work that went into crosses, data collection, and analysis. The manuscript involves a thoughtful analysis of how different the genetic architecture under selection was depending on type of selection and nutrient content.

While the manuscript is generally easy to read, it is quite long and I found myself struggling to maintain comprehension of all the sections of the results as I read the manuscript. I wonder if it would be worthwhile to try to reduce and combine some of the sections of the results to make the whole of this large body of work more digestible for the average reader. That said, I do think all of the content for this manuscript should be published as a single unit, as it would also be difficult to understand each of the pieces of the study in isolation.

Additional comments:

There were a number of minor typos (e.g. extra periods at the ends of sentences) throughout the manuscript. Make sure to correct them with revisions.

22: “where previous association mapping failed to recover much of its heritability” is confusing; perhaps “where loci identified by association mapping failed to explain heritability estimates”

188: If median absolute deviation is equivalent to standard deviation, why not use standard deviation?

188, 259 407: it may be better to say “analogous” or “similar” instead of “equivalent” here, since standard deviation and median absolute deviation will give different values

261-262: I am a bit confused as to why drift is being equated with random selection here rather than being defined as a sampling error based on population size

Fig. S6: The threshold here looks the same as the threshold used for directionally selected populations, yet stabilizing selection resulted in higher than expected heterozygosity. Shouldn’t this figure use a different threshold calculated from populations under stabilizing selection? A few peaks (chr 1 LN B and C, chr 4 HN A and B) look like they may be close, and an overly stringent threshold could falsely indicate that there was no detectable effect of selection (line 487) on heterozygosity in populations under stabilizing selection.

Fig. 6A: righthand side (chromosome numbers?) got cut off

516: On first reading, this sentence appears to contradict the one on line 513. Perhaps “not biased towards any one particular accession haplotype among different selective sweeps”

Fig. S7C: What do the black points indicate?

Fig. S10Aiii, Biii: It would be easier to compare these two panels if the x-axes covered the same range.

723: This is a serendipitous thing that happened and it brings me great joy that the authors decided to explore it.

785: It might be good to clarify the statement “However, most of the genetic variation was not recovered.” to make it clear that loci identified in the study only explained a fraction of the genetic variation in the trait.

815: I don’t understand this sentence- what is the robustness of branching traits?

823: I was not previously aware of the Jinks-Connolly rule. This is an interesting point. Also, make sure to spell Connolly correctly in the revision.

854: Please clarify what is meant by “axillary meristem formation.” Do you mean initiation of axillary branches from axillary buds?

882-885: How much do you think that this being a MAGIC population with such a large number of founder alleles lead to this result of different loci being selected upon in different populations? It seems like the level of standing genetic variation and low allele frequencies at causative loci would not be very representative of “natural” Arabidopsis populations. I realize that mimicking natural populations was not the goal here, but it might be worth writing some reflections on how the MAGIC design itself influenced the results. The authors do do this on line 966-969, but they may want to make it more explicit here as well or instead.

**Have all data underlying the figures and results presented in the manuscript been provided?**

Reviewer #1: Yes

Reviewer #2: Yes

Reviewer #3: Yes

Reviewer #4: Yes

PLOS authors have the option to publish the peer review history of their article (what does this mean?). If published, this will include your full peer review and any attached files.

Reviewer #1: No

Reviewer #2: No

Reviewer #3: No

Reviewer #4: No

---

## [Decision Letter · Decision Letter 1]

6 Jun 2023

Dear Dr Leyser,

Thank you very much for submitting your Research Article entitled 'Artificial selection reveals complex genetic architecture of shoot branching and its response to nitrate supply in Arabidopsis' to PLOS Genetics.

The manuscript was fully evaluated at the editorial level and by independent peer reviewers. The editors and the reviewers appreciated the attention to an important topic but identified some concerns that we ask you address in a revised manuscript.

We therefore ask you to modify the manuscript according to the review recommendations. Your revisions should address the specific points made by each reviewer.

Yours sincerely,

Magnus Nordborg

Guest Editor

PLOS Genetics

Claudia Köhler

Section Editor

PLOS Genetics

**Comments of the guest editor:**

The manuscript is clearly massively improved, and I see no reason not to publish. As noted by Reviewer 2, there are still several things one could quibble about. These, I believe, mostly stem from writing a paper in a different field with a huge literature. That said, the reviewer is right, and reading a few more papers wouldn't hurt. But I leave this up to you. There are many such examples, e.g. the notion of associative overdominance, which you seemingly attribute to a 1998 paper, but which I'm confident is far older (the 1998 paper cites Frydenberg, 1963, but I guess the notion is in Fisher somewhere). Anyway, we're not doing history of science here, and the readers can figure it out.

On slight thing I would fix is l. 41, "[o]ur results suggest that this trait is controlled by tens of loci of small effect" — you mean "a minium...". The bottom line is that you have limited power to detect really small effects, and that your resolution is limited by haplotype structure and linkage disequilibrium, which means that your sentence if by "loci" you mean QTL, e.g. large haplotypes with unknown numbers of causative polymorphisms.

Reviewer's Responses to Questions

**Comments to the Authors:**

Reviewer #1: The authors have addressed my comments and the revised manuscript is easier to follow.

Reviewer #2: First, I would like to thank the authors for the careful consideration given to my comments and those of the other reviewers. I am convinced by their arguments that there is little signal of parallel change across their selection populations. This supports their heterozygosity analysis as a useful summary of results.

The authors argue that genetic redundancy led to a genetically heterogenous response to selection. Given this conclusion, my only advice is to broaden the discussion of genetic redundancy. In truth, the empirical evidence for genetic redundancy in quantitative traits is NOT “extremely sparse.” Redundancy is built directly into most multi-locus genetic models. It is supported by multiple lines of evidence for quantitative traits in many systems (see summaries in Falconer and Mackay or Lynch and Walsh). For a nice example in the specific context of selection experiments, the authors might revisit several papers from F.M. Cohan published in Evolution around 1984 (in D. melanogaster), but there are many other examples.

Reviewer #3: The authors have accounted for my comments and those of the associate editor and of other reviewers and the paper can now been published.

**Have all data underlying the figures and results presented in the manuscript been provided?**

Reviewer #1: Yes

Reviewer #2: None

Reviewer #3: Yes

PLOS authors have the option to publish the peer review history of their article (what does this mean?). If published, this will include your full peer review and any attached files.

Reviewer #1: No

Reviewer #2: No

Reviewer #3: No

---

## [Editor Report · Decision Letter 2]

8 Jul 2023

Dear Dr Leyser,

We are pleased to inform you that your manuscript entitled "Artificial selection reveals complex genetic architecture of shoot branching and its response to nitrate supply in Arabidopsis" has been editorially accepted for publication in PLOS Genetics. Congratulations!

Yours sincerely,

Magnus Nordborg

Guest Editor

PLOS Genetics

Claudia Köhler

Section Editor

PLOS Genetics

Comments from the reviewers (if applicable):

**Data Deposition**

http://datadryad.org/submit?journalID=pgenetics&manu=PGENETICS-D-22-00721R2

**Press Queries**

---

## [Editor Report · Acceptance letter]

18 Aug 2023

PGENETICS-D-22-00721R2 

Artificial selection reveals complex genetic architecture of shoot branching and its response to nitrate supply in Arabidopsis 

Dear Dr Leyser, 

We are pleased to inform you that your manuscript entitled "Artificial selection reveals complex genetic architecture of shoot branching and its response to nitrate supply in Arabidopsis" has been formally accepted for publication in PLOS Genetics! Your manuscript is now with our production department and you will be notified of the publication date in due course.

With kind regards,

Lilla Horvath

PLOS Genetics

On behalf of:
